# SPATIALLY STRUCTURED RECURRENT MODULES

**Nasim Rahaman**[1,2]  **Anirudh Goyal**[2]  **Muhammad Waleed Gondal**[1]  **Manuel Wuthrich**[1]

**Stefan Bauer**[1]  **Yash Sharma**[3]  **Yoshua Bengio**[2,4]  **Bernhard Schölkopf**[1]

## ABSTRACT

Capturing the structure of a data-generating process by means of appropriate inductive biases can help in learning models that generalize well and are robust to changes in the input distribution. While methods that harness spatial and temporal structures find broad application, recent work has demonstrated the potential of models that leverage sparse and modular structure using an ensemble of sparingly interacting modules. In this work, we take a step towards dynamic models that are capable of simultaneously exploiting both modular and spatiotemporal structures. To this end, we model the dynamical system as a collection of autonomous but sparsely interacting *sub-systems* that interact according to a learned topology which is informed by the spatial structure of the underlying system. This gives rise to a class of models that are well suited for capturing the dynamics of systems that only offer *local views* into their state, along with corresponding spatial locations of those views. On the tasks of video prediction from cropped frames and multi-agent world modeling from partial observations in the challenging Starcraft2 domain, we find our models to be more robust to the number of available views and capable of better generalization to novel tasks without additional training than strong baselines that perform equally well or better on the training distribution.

## 1 INTRODUCTION

Many spatiotemporal complex systems can be abstracted as a collection of autonomous but sparsely interacting sub-systems, where sub-systems tend to interact if they are in each others' *vicinity*. As an illustrative example, consider a grid of traffic intersections. Traffic flows from a given intersection to the adjacent ones, and the actions taken by some "agent", say an autonomous vehicle, may at first only affect its immediate surroundings.

Now suppose we want to forecast the future state of the traffic grid (say for the purpose of avoiding traffic jams). There is a spectrum of possible strategies for modeling the system at hand. On one extreme lies the most general strategy which considers the entirety of all intersections simultaneously to predict the next state of the grid (Figure 1c). The resulting model class can in principle account for interactions between any two intersections, irrespective of their spatial distance. However, the number of interactions such models must consider does not scale well with the size of the grid, and this strategy might be rendered infeasible for large grids with hundreds of intersections. On the other end of the spectrum is a strategy which abstracts the dynamics of each intersection as an autonomous *sub-system*, with each sub-system interacting only with its immediate neighbors (Figure 1a). The interactions may manifest as messages that one sub-system passes to another and possibly contain information about how many vehicles are headed towards which direction, resulting in a collection of message passing entities (i.e. sub-systems) that collectively model the entire grid. By adopting this strategy, one assumes that the immediate future of any given intersection is affected only by the present states of the neighboring intersections, and not some intersection at the opposite end of the grid. The resulting class of models scales well with the size of the grid, but is possibly unable to model certain long-range interactions that could be leveraged to efficiently distribute traffic flow.

The spectrum above parameterizes the extent to which the spatial structure of the underlying system informs the design of the model. The former extreme ignores spatial structure altogether, resulting

---

[1]Max-Planck Institute for Intelligent Systems Tübingen, [2]Mila, Québec, [3]Bethgelab, Eberhard Karls Universität Tübingen, [4]Université de Montreal. Correspondence to: <nasim.rahaman@tuebingen.mpg.de>.

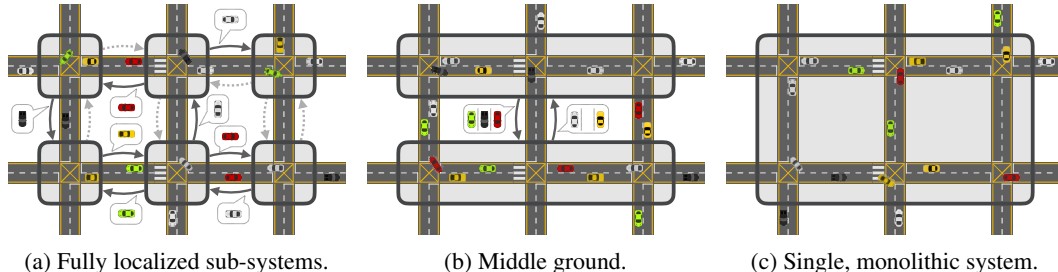

(a) Fully localized sub-systems.  (b) Middle ground.  (c) Single, monolithic system.

Figure 1: A schematic representation of the spectrum of modeling strategies. Solid arrows with speech bubbles denote (dynamic) messages being passed between *sub-systems* (dotted arrows denote the lack thereof). **Gist:** on one end of the spectrum, (Figure 1a), we have the strategy of abstracting each intersection as a sub-system that interact with neighboring sub-systems. On the other end of the spectrum (Figure 1c) we have the strategy of modeling the entire grid with one monolithic system. The middle ground (Figure 1b) we explore involves letting the model *develop* a notion of locality by (say) abstracting entire avenues with a single sub-system.

in a class of models that can be expressive but whose sample and computational complexity do not scale well with the size of the system. The latter extreme results in a class of models that can scale well, but its adequacy (in terms of expressivity) is contingent on a predefined notion of locality (in the example above: the immediate four-neighborhood of an intersection). In this work, we aim to explore a middle-ground between the two extremes: namely, by proposing a class of models that *learns* a notion of locality instead of relying on a predefined one (Figure 1b). Reconsidering the traffic grid example: the proposed strategy results in a model that may learn to abstract (say) entire avenues with a single sub-system, if it is useful towards solving the prediction task. This yields a scheme where a single sub-system might account for events that are spatially distant (such as those in the opposite ends of an avenue), while events that are spatially closer together (like those on two adjacent avenues of the same street, where streets run perpendicular to avenues) might be assigned to different sub-systems.

To implement this scheme, we build on a framework wherein the sub-systems are modelled as independent recurrent neural network (RNN) modules that interact sparsely via a bottleneck of attention (a variant of which is explored in Goyal et al. (2019)) while extending it along two salient dimensions. First, we learn an interaction topology between the sub-systems, instead of assuming that all sub-systems interact with all others in an all-to-all topology. We achieve this by learning to embed each sub-system in a space endowed with a metric, and attenuate the interaction between two given sub-systems according to their distance in this space (i.e., sub-systems too far away from each other in this space are not allowed to interact). Second, we relax a common assumption that the entire system is perceived simultaneously; instead, we only assume access to *local* (partial) observations alongside with the associated *spatial locations*, resulting in a setting that partially resembles that of Eslami et al. (2018). Expressed in the language of the example above: we do not expect a bird's eye view of the traffic grid, but only (say) LIDAR observations from autonomous vehicles at known GPS coordinates, or video streams from traffic cameras at known locations. The spatial location associated with an observation plays a crucial role in the proposed architecture in that we map it to the embedding space of sub-systems and *address* the corresponding observation only to sub-systems whose embeddings lie in close vicinity. Likewise, to predict future observations at a queried spatial location, we again map said location to the embedding space and poll the states of sub-systems situated nearby. The result is a model that can *learn* which spatial locations are to be associated with each other and be accounted for by the same sub-system. As an added plus, the parameterization we obtain is not only agnostic to the number of available observations and query locations, but also to the number of sub-systems.

To evaluate the proposed model, we choose a problem setting where **(a)** the task is composed of different sub-systems or *processes* that locally interact both spatially and temporally, and **(b)** the environment offers local views into its state paired with their corresponding spatial locations. The challenge here lies in building and maintaining a consistent representation of the *global state* of the system given only a set of partial observations. To succeed, a model must learn to efficiently capture the available observations and place them in an appropriate spatial context. The first problem we consider is that of video prediction from crops, analogous to that faced by visual systems of many animals: given a set of small crops of the video frames centered around stochastically sampled pixels (corresponding to where the fovea is focused), the task is to predict the content of a crop

around *any* queried pixel position at a future time. The second problem is that of multi-agent world modeling from partial observations in spatial domains, such as the challenging Starcraft2 domain (Samvelyan et al., 2019; Vinyals et al., 2017). The task here is to model the dynamics of the *global* state of the environment given *local* observations made by cooperating agents and their corresponding actions. Finally, we also include visualizations on a multi-agent grid-world environment designed for simulating railroad traffic (Eichenberger et al., 2019). Importantly and unlike prior work (Sun et al., 2019), our parameterization is agnostic to the number of agents in the environment, which can be flexibly adjusted on the fly as new agents become available or existing agents retire. This is beneficial for generalization in settings where the number of agents during training and testing are different.

**Contributions. (a)** We propose a new class of models, which we call Spatially Structured Recurrent Modules or S2RMs, which perform attention-driven modular computations according to a learned spatial topology. **(b)** We evaluate S2RMs (along with several strong baselines) on a selection of challenging problems and find that S2RMs are robust to the number of available observations and can generalize to novel tasks.

## 2 PROBLEM STATEMENT

In this section, we build on the intuition from the previous section to formally specify the problem we aim to approach with the methods described in the later sections. To that end, let $\mathcal{X}$ be a metric space, $\mathcal{O}$ some set of possible *observations*, and $\mathfrak{O}_{\mathcal{X}}$ a set of mappings $\mathcal{X} \to \mathcal{O}$. Now, consider the *evolution function* of a discrete-time dynamical system:

$$\phi : \mathbb{Z} \times \mathfrak{O}_{\mathcal{X}} \to \mathfrak{O}_{\mathcal{X}} \text{ satisfying:} \tag{1}$$
$$\phi(0, \mathfrak{o}) = \mathfrak{o} \text{ where } \mathfrak{o} \in \mathfrak{O}_{\mathcal{X}} \quad \text{and} \quad \phi(t_2, \phi(t_1, \mathfrak{o})) = \phi(t_1 + t_2, \mathfrak{o}) \text{ for } t_1, t_2 \in \mathbb{Z}$$

Informally, $\mathfrak{o}$ can be interpreted as the *world state* of the system; together with a spatial *location* $\mathbf{x} \in \mathcal{X}$, it gives the *local* observation $\mathbf{O} = \mathfrak{o}(\mathbf{x}) \in \mathcal{O}$. Given an initial world state $\mathfrak{o}$, the mapping $\phi(t, \mathfrak{o})$ yields the world state $\mathfrak{o}_t$ at some (future) time $t$, thereby characterizing the dynamics of the system (which might be stochastic). The problem we consider is the following:

**Problem:** At every time step $t = 0, ..., T$, we are given a set of positions $\{\mathbf{x}_t^a\}_{a=1}^A$ and the corresponding observations $\{\mathbf{O}_t^a\}_{a=1}^A$, where $\mathbf{O}_t^a := \mathfrak{o}_t(\mathbf{x}^a)$. The task is to infer the world state $\mathfrak{o}_{t'}$ at some future time-step $t' > T$ in order to predict $\mathbf{O}_{t'}^q = \mathfrak{o}_{t'}(\mathbf{x}^q)$ at some arbitrary query position $\mathbf{x}^q$.

In the traffic grid example of Section 1, one could imagine $a$ as indexing traffic cameras or autonomous vehicles (i.e., *observers*), $\mathbf{x}_t^a$ as the GPS coordinates of observer $a$, and $\mathbf{O}_t^a$ as the corresponding sensor feed (e.g. LIDAR observations or video streams from vehicles or traffic cameras).

## 3 MODELLING ASSUMPTIONS

Given the problem in Section 2, we now make certain modelling assumptions. These assumptions will ultimately inform the inductive biases we select for the model (proposed in Section 4); nevertheless, we remark beforehand that as with any inductive bias, their applicability is subject to the properties of the system being modeled and the objectives being optimized (OOD generalization, etc).

**Recurrent Dynamics Modeling.** While there exist multiple ways of modeling dynamical systems, we shall focus on recurrent neural networks (RNNs). Typically, RNN-based dynamics models are expressed as functions of the form:

$$\mathbf{h}_{t+1} = F(\mathbf{O}_t, \mathbf{h}_t) \qquad \mathbf{O}_t = D(\mathbf{h}_t) \tag{2}$$

where $\mathbf{O}_t$ is the observation at time $t \in \mathbb{Z}$, and $\mathbf{h}_{t+1}$ is the hidden state of the model. $F$ can be thought of as the parameterized *forward-evolution function* the hidden state $\mathbf{h}$ conditioned on the observation $\mathbf{O}$, whereas $D$ is a *decoder* that maps the hidden state to observations.

**Decomposition into Locally Interacting Sub-systems.** We make the assumption that the dynamical system $\phi$ can be decomposed to constituent sub-systems $(\phi_1, \phi_2, ..., \phi_M)$ that dynamically and sparsely interact with each other while respecting some *interaction topology*. By interaction topology, we mean that each module $\phi_i$ can be identified with an embedding $\mathbf{p}_i$ in a topological space $\mathcal{S}$ equipped with a similarity kernel $Z$ and that the sub-system $\phi_i$ may preferentially interact with another sub-system $\phi_j$ if their respective embeddings are close in $\mathcal{S}$ with respect to $Z$, i.e. if

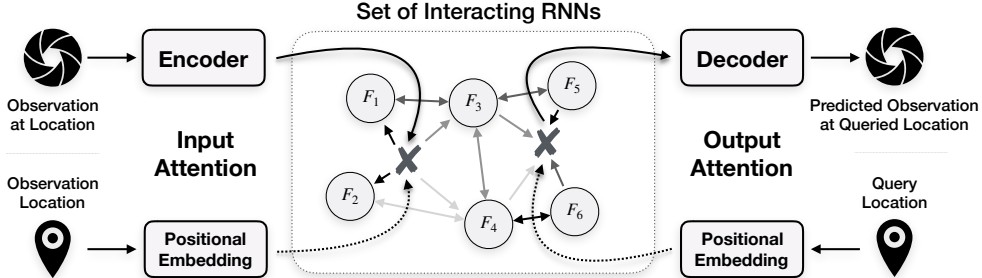

Figure 2: Schematic illustration of the proposed architecture. An observation is addressed to modules with embeddings situated in vicinity of its embedded location. Likewise, modules with embeddings in the vicinity of an embedded query location are polled to produce a prediction.

$Z(\mathbf{p}_i, \mathbf{p}_j)$ is large. Intuitively, one may think of $Z$ as inducing a notion of locality between sub-systems, according to which $\phi_j$ lies in the *local vicinity* of $\phi_i$.

**Locality of Observations.** The notion of locality between sub-systems induced by $Z$ is distinct from that induced by the metric of space $\mathcal{X}$ of locations in the environment (cf. Section 2), and one important modelling decision is how these two should interact. We propose to embed the position $\mathbf{x} \in \mathcal{X}$ associated with an observation $\mathbf{O}$ to the metric space of sub-systems $\mathcal{S}$ via a continuous[1] and injective mapping $P : \mathcal{X} \to \mathcal{S}$. This allows us to *match* the observation $\mathbf{O}$ to all sub-systems $\phi_m$ that are in the vicinity of $P(\mathbf{x}) \in \mathcal{S}$, i.e., where $Z(P(\mathbf{x}), \mathbf{p}_m)$ is sufficiently large. Each subsystem $\phi_m$ therefore accounts for observations made at a set of positions $\mathcal{X}_m \subset \mathcal{X}$, which we call its *enclave*.

## 4 SPATIALLY STRUCTURED RECURRENT MODULES (S2RMS)

Informed[2] by the model assumptions detailed in the previous section, we now describe the proposed model (Figure 2) comprising the following components:

**Model Inputs.** Recall from Section 2 that we have for every time step $t = 0, ..., T$ a set of tuples of positions and observations $\{(\mathbf{x}_t^a, \mathbf{O}_t^a)\}_{a=1}^A$ where $\mathbf{x}_t^a \in \mathcal{X}$ and $\mathbf{O}_t^a \in \mathcal{O}$ for all $t$ and $a$. To simplify, we assume that $\mathcal{X} \subset \mathbb{R}^n$, and denote by $x_i$ the $i$-th component of the vector $\mathbf{x} \in \mathcal{X}$.

**Encoder.** The encoder $E$ is a parameterized function mapping observations $\mathbf{O}$ to a corresponding vector representation $\mathbf{e} = E(\mathbf{O})$. Here, $E$ processes all observations in parallel across $t$ and $a$ to yield representations $\mathbf{e}_t^a$.

**Positional Embedding.** The positional embedding $P$ is a fixed mapping from $\mathcal{X}$ to $\mathcal{S}$. We choose $\mathcal{S}$ to be the unit sphere in $d$-dimensions, $d$ being a multiple of $2n$, and the positional encoder as the following function:

$$P(\mathbf{x}) = \mathbf{s}/\|\mathbf{s}\| \in \mathcal{S} \quad \text{where} \quad (s_{i+m}, s_{i+1+m}) = (\sin(x_m/10000^i), \cos(x_m/10000^i)) \tag{3}$$

with $m = 0, ..., n-1$ and $i = 0, 2, ..., d/n - 1$. The above function finds common use (Vaswani et al., 2017; Mildenhall et al., 2020; Zhong et al., 2020) and can be motivated from the perspective of Reproducing Kernel Hilbert Spaces (Rahimi & Recht, 2007) (see Appendix C.3 for a discussion). We henceforth refer to $P(\mathbf{x})$ as $\mathbf{s}$ and $P(\mathbf{x}_t^a)$ as $\mathbf{s}_t^a$.

**Set of Interacting RNNs.** To model the dynamics of the world state, we use a set of $M$ independent RNN *modules*, which we denote as $\{F_m\}_{m=1}^M$. To each $F_m$, we associate an embedding vector $\mathbf{p}^m \in \mathcal{S}$, where all $\{\mathbf{p}^m\}_{m=1}^M$ are learnable parameters. The RNNs $F_m$ interact with each other via an *inter-cell attention*, and with the input representations $\mathbf{e}_t^a$ via *input attention*. Precisely, at a given time step $t$, each $F_m$ expects an input $\mathbf{u}_t^m$, an *aggregated hidden state* $\bar{\mathbf{h}}_t^m$ and optionally, a memory state $\mathbf{c}_t^m$ to yield the hidden and memory states at the next time step:

$$(\mathbf{h}_{t+1}^m, \mathbf{c}_{t+1}^m) = F_m(\mathbf{u}_t^m, \bar{\mathbf{h}}_t^m, \mathbf{c}_t^m) \tag{4}$$

---

[1]The continuity of $P$ ties the two notions of locality by requiring that an infinitesimal change in $\mathbf{x}$ corresponds to one in $\mathcal{S}$. Injectivity ensures that no two points in $\mathcal{X}$ are mapped to the same point in $\mathcal{S}$.

[2]In doing so, we use the assumptions merely as guiding principles; we do not claim that we *infer* the true decomposition of the ground-truth system, even if all assumptions are satisfied.

where the input $\mathbf{u}_t^m$ results from the input attention and $\bar{\mathbf{h}}_t^m$ from the inter-cell attention (see below).

**Kernel Modulated Dot-Product Attention.** A central component of the proposed architecture is the kernel modulated dot-product attention (KMDPA), which we now define. First, we let $Z : \mathcal{S} \times \mathcal{S} \to [0, 1]$ be the following kernel:

$$Z(\mathbf{p}, \mathbf{s}) = \begin{cases} \exp\left[-2\epsilon(1 - \mathbf{p} \cdot \mathbf{s})\right], & \text{if } \mathbf{p} \cdot \mathbf{s} \geq \tau \\ 0, & \text{otherwise} \end{cases} \tag{5}$$

where $\epsilon \in (0, \infty)$ is the kernel bandwidth, and $\tau \in [-1, 1)$ is the truncation parameter (additional details in Appendix C.1). Now, KMDPA maps two sets $\mathbf{A}$ and $\mathbf{B}$ to a third set $\hat{\mathbf{A}}$, where:

$$\mathbf{A} = \{(\mathbf{a}^i, \mathbf{y}^i)\}_{i=1}^I; \quad \mathbf{B} = \{(\mathbf{b}^j, \mathbf{z}^j)\}_{j=1}^J; \quad \hat{\mathbf{A}} = \{(\mathbf{a}^k, \hat{\mathbf{y}}^k)\}_{k=1}^I = \text{KMDPA}(\mathbf{A}, \mathbf{B}) \tag{6}$$

Here, $\mathbf{a}^i, \mathbf{b}^j \in \mathcal{S}$, and $\mathbf{y}^i, \mathbf{z}^j$ are vectors of not necessarily the same dimension. In order to evaluate $\hat{\mathbf{A}}$, we first compute the interaction weights $W_{ij}$ between any two pairs of entities $(\mathbf{a}^i, \mathbf{y}^i)$ and $(\mathbf{b}^j, \mathbf{z}^j)$, which depends on a *local term* $W_{ij}^{(L)}$ and a *non-local term* $\bar{W}_{ij}$. We have:

$$\bar{W}_{ij} = \text{softmax}_j\left(\Theta^{(\text{Query})}(\mathbf{y}^i) \cdot \Theta^{(\text{Key})}(\mathbf{z}^j)\right); \quad W_{ij}^{(L)} = Z(\mathbf{a}^i, \mathbf{b}^j); \quad W_{ij} = \bar{W}_{ij} W_{ij}^{(L)} \tag{7}$$

where $\Theta^{(\text{Query})}$ and $\Theta^{(\text{Key})}$ are learnable linear mappings that project $\mathbf{y}^i$ and $\mathbf{z}^j$ to the same space. The penultimate step computes the following two quantities:

$$\tilde{\mathbf{y}}^i = \sum_j W_{ij}^{(L)} \mathbf{y}^j; \qquad \bar{\mathbf{y}}^i = \sum_j \bar{W}_{ij} \Theta^{(\text{Value})}(\mathbf{z}^j) \tag{8}$$

where $\Theta^{(\text{Value})}$ is another learnable linear function mapping from the vector space of $\mathbf{z}$ to that of $\mathbf{y}$. Finally, we have:

$$\hat{\mathbf{y}}^i = G(\tilde{\mathbf{y}}^i, \bar{\mathbf{y}}^i) \cdot \tilde{\mathbf{y}}^i + \left(1 - G(\tilde{\mathbf{y}}^i, \bar{\mathbf{y}}^i)\right) \cdot \bar{\mathbf{y}}^i \tag{9}$$

where $G$ is a gating layer with sigmoid non-linearity implementing a soft selection mechanism between the linear combination of values ($\bar{\mathbf{y}}^i$) and the inputs weighted by the local weights ($\tilde{\mathbf{y}}^i$). In what follows, we will refer to the set $\mathbf{A}$ as *query set*, $\mathbf{B}$ as *key set* and $\hat{\mathbf{A}}$ as *output set*.

**Input Attention.** The input attention mechanism is a KMDPA, mapping between sets of observation tuples $\{(\mathbf{s}_t^a, \mathbf{e}_t^a)\}_{a=1}^A$ (key set) and the current RNN-states $\{(\mathbf{p}^m, \mathbf{h}_t^m)\}_{m=1}^M$ (query set) to that of RNN inputs $\{(\mathbf{p}^n, \mathbf{u}_t^n)\}_{n=1}^M$ (output set). Now on the one hand, we observe that the input $u_t^m$ to RNN $F_m$ can contain information about an observations $\mathbf{e}_t^a$ only if the embedded location of the said observation $\mathbf{s}_t^a$ is close enough to the embedding of the RNN $\mathbf{p}^m$ in $\mathcal{S}$ (i.e. if $Z(\mathbf{s}_t^a, \mathbf{p}^m) > 0$), thereby implementing the assumption of *locality of observations*. On the other hand, the non-local term allows a module $F_m$ to reject (or accept) an observation based on its content, which can be beneficial if two modules attend to overlapping regions in the environment but specialize to different aspects of the dynamics. Please refer to Appendix C.1 for a precise description of the mechanism, in particular the (optional) use of multiple dot-product attention heads.

**Inter-cell Attention.** The intercell attention mechanism is another KMDPA, mapping two copies of the current RNN-states $\{(\mathbf{p}^m, \mathbf{h}_t^m)\}_{m=1}^M$ (one as query and another as key set) to the set of *aggregated hidden states* $\{(\mathbf{p}^n, \bar{\mathbf{h}}_t^n)\}_{n=1}^M$ (output set). This enables local interaction between the RNNs $F_m$, in that the local term ensures that RNN $F_m$ interacts with RNN $F_n$ only if their respective embeddings $\mathbf{p}^m$ and $\mathbf{p}^n$ are close enough in $\mathcal{S}$ (i.e. if $Z(\mathbf{p}^m, \mathbf{p}^n) > 0$), thereby implementing the assumption of *local interactions between sub-systems*. The non-local term allows two modules to interact with each other based on their hidden states, i.e. it provides the mechanism for a module to (not) interact with another other based on their respective states, even if their embeddings are similar enough in $\mathcal{S}$. Appendix C.2 contains a precise description of the attention mechanism.

**Output Attention.** The output attention mechanism together with the decoder (described below) serve as an apparatus to evaluate the world state modeled implicitly by the set of RNNs $(\{F_m\}_{m=1}^M)$ at time $t + 1$ (for one-step forward models). Given a query location $\mathbf{x}^q \in \mathcal{X}$ and its corresponding embedding $\mathbf{s}^q$, the output attention mechanism polls the RNNs $F_m$ whose embeddings $\mathbf{p}^m$ are similar enough to $\mathbf{s}^q$, as measured by the kernel $Z$. Denoting by $h_{mj}$ the $j$-th component of $\mathbf{h}_{t+1}^m$ and by $d_j^q$ the $j$-th component of the vector $\mathbf{d}_{t+1}^q$ associated with the query location $\mathbf{x}^q$, we have:

$$d_j^q = \sum_m Z(\mathbf{s}^q, \mathbf{p}^m) \, h_{mj} \tag{10}$$

**Decoder.** The decoder $D$ is a parameterized function that predicts the observation $\hat{\mathbf{O}}_{t+1}^q \in \mathcal{O}$ at $\mathbf{x}^q$ given the representation $\mathbf{d}_{t+1}^q$ from the output attention.

This concludes the description of the generic architecture, which allows for flexibility in the choice of the RNN architecture (i.e., the internal architecture of $F_m$). In practice, we find Gated Recurrent Units (GRUs) (Cho et al., 2014) to work well, and call the resulting model Spatially Structured GRU or **S2GRU**. Moreover, Relational Memory Cores (RMCs) (Santoro et al., 2018) also profit from our architecture (with a modification detailed in Appendix E.3), and we call the resulting model **S2RMC**.

## 5 RELATED WORK

**Problem Setting.** Recall that the problem setting we consider is one where the environment offers local (partial) views into its global state paired with the corresponding spatial locations. With Generative Query Networks (GQNs), Eslami et al. (2018) investigate a similar setting where the 2D images of 3D scenes are paired with the corresponding *viewpoint* (camera position, yaw, pitch and roll). Given that GQNs are feedforward models, they do not consider the dynamics of the underlying scene and as such cannot be expected to be consistent over time (Kumar et al., 2018). Singh et al. (2019) and Kumar et al. (2018) propose variants that are temporally consistent, but unlike us, they do not focus on the problem of predicting the future state of the system.

**Modularity.** Modularity has been a recurring topic in the context of meta-learning (Alet et al., 2018; Bengio et al., 2019; Ke et al., 2019), sequence modeling (Ghahramani & Jordan, 1996; Henaff et al., 2016; Li et al., 2018; Goyal et al., 2019; Mei et al., 2020; Mittal et al., 2020) and beyond (Jacobs et al., 1991; Shazeer et al., 2017; Parascandolo et al., 2017). In the context of RNNs, Li et al. (2018) explore a setting where the recurrent units operate entirely independently of each other. Closer to our work, Goyal et al. (2019) explores the setting where autonomous RNN modules interact with each other via the bottleneck of sparse attention. However, instead of leveraging the spatial structure of the environment, they induce sparsity using a scheme inspired by the k-winners-take-all principle (Majani et al., 1988) where only the $k$ modules that attend the most to the input are activated and propagate their state forward, whereas the remaining modules that do not receive an input follow *default dynamics* in that their hidden states are not updated. This can be contrasted with S2RMs, where the modules that do not receive inputs may still evolve their states forward in time, reflecting that the environment may evolve even when no observations are available.

**Attention Mechanisms and Information Flow.** Attention mechanisms have been used to attenuate the flow of information between components of the network, e.g. (Graves et al., 2014; 2016; Santoro et al., 2018; Ke et al., 2018; Veličković et al., 2017; Battaglia et al., 2018). There is a growing interest in efficient attention mechanisms for use in transformers (Vaswani et al., 2017), and like KMDPA, some recently proposed methods rely on learned sparsity (Kitaev et al., 2020; Tay et al., 2020). However, these induce sparsity by dynamically clustering or sorting based on content, while we make explicit use of the spatial information accompanying observations to learn a spatially-grounded sparsity pattern. Moreover, mechanisms for spatial attention have also been studied (Jaderberg et al., 2015; Wang et al., 2017; Zhang et al., 2018; Parmar et al., 2018), but they typically operate on image pixels. Our setting is different in that we do not assume that the world-state (from which we sample local observations) can be represented as an image.

## 6 EXPERIMENTS

In this section, we present a selection of experiments to quantitatively evaluate S2RMs and gauge their performance against strong baselines on two data domains, namely video prediction from crops on the well-known bouncing-balls domain and multi-agent world modelling from partial observations in the challenging Starcraft2 domain. We also include qualitative visualizations on a grid-world task in Appendix A. Additional tables, results and supporting plots can be found in Appendix F.

**Baselines.** To draw fair comparisons between various RNN architectures, we require an *architectural scaffolding* that is agnostic to the number of observations $A$, is invariant to the ordering of $\{(\mathbf{x}_t^a, \mathbf{O}_t^a)\}_a^A$ with respect to $a$ and features a querying mechanism to extract a predicted observation $\mathbf{O}_{t'}^q$ at a given query location $\mathbf{x}^q$ in a future time-step $t' > t$. Fortunately, it is possible to obtain a performant class of models fulfilling our requirements by extending prior work on Generative Query Networks or GQNs (Eslami et al., 2018). The resulting model has three components: an encoder,

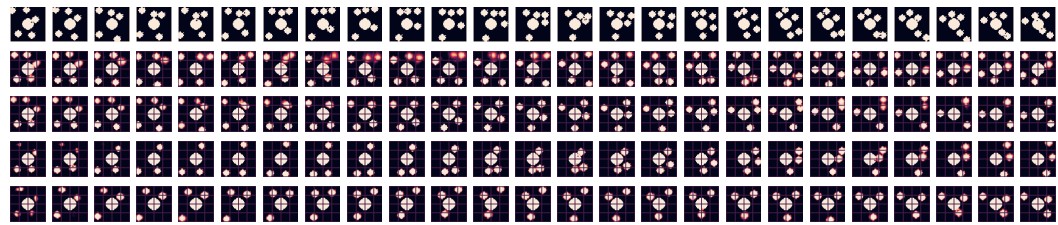

Figure 3: Rollouts (OOD) with 5 bouncing balls, from top to bottom: ground-truth, S2GRU, RIMs, RMC, LSTM. Note that all models were trained on sequences with 3 bouncing balls, and the global state is reconstructed by stitching together 16 patches of size $11 \times 11$ produced by the models (queried on a $4 \times 4$ grid). **Gist:** S2GRU succeeds at keeping track of all bouncing balls over long rollout horizons (25 frames).

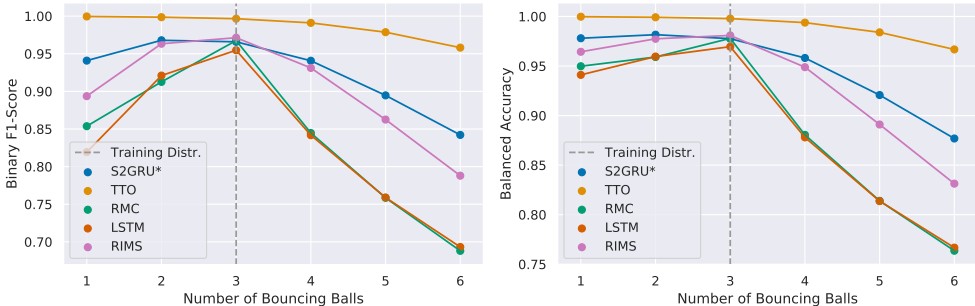

Figure 4: Performance metrics on OOD one-step forward prediction task. **Gist:** S2GRU outperforms all RNN baselines OOD.

a RNN, and a decoder, which we describe in detail in Appendix D. In our experiments, we fix the encoder and decoder to be essentially identical to those in S2RMs, but vary the architecture of the RNN, where we experiment with LSTMs (Hochreiter & Schmidhuber, 1997), RMCs (Santoro et al., 2018) and RIMs (Goyal et al., 2019). As a sanity check, we also show results with a *Time Travelling Oracle* (TTO), which at time-step $t$ has access to the (partially observed) state at $t + 1$. Its purpose is to verify that the architectural scaffolding around the baseline RNNs (defined in Appendix D) does not constrain their performance and that the comparison to S2RMs is indeed fair.

**Video Prediction from Crops.** We consider the problem of predicting the future frames of simulated videos of balls bouncing in a closed box, given only crops from the past video frames which are centered at known pixel positions. Using the notation introduced in Section 2: at every time step $t$, we sample $A = 10$ pixel positions $\{\mathbf{x}_t^a\}_{a=1}^{10}$ from the $t$-th full video frame $\mathfrak{o}_t$ of size $48 \times 48$. Around the sampled central pixel positions $\mathbf{x}_t^a$, we extract $11 \times 11$ crops, which we use as the *local* observations $\mathbf{O}_t^a$. The task now is to predict $11 \times 11$ crops $\mathbf{O}_{t'}^q$ corresponding to query central-pixel-positions $\mathbf{x}_{t'}^q$ at a future time-step $t' > t$. Observe that at any given time-step $t$, the model has access to at most 52% of the global video frame assuming that the crops never overlap (which is rather unlikely).

Having trained on the training dataset with 3 bouncing balls, we evaluate the forward-prediction performance on all test datasets with 1 to 6 bouncing balls. Given that we treat the prediction problem as a pixel-wise binary classification problem, we report the balanced accuracy (i.e. arithmetic mean of recall and specificity) or F1-scores (i.e. harmonic mean of precision and recall) to account for class-imbalance. In Figure 4, we see that S2GRUs out-perform

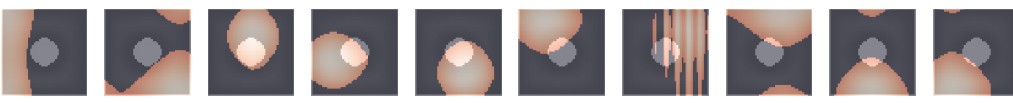

Figure 5: Visualization of the spatial locations each module is responsible for modeling (i.e. the *enclaves* $\mathcal{X}_m$, defined in Section 3). The central ball does not bounce, i.e. it is stationary in all sequences. **Gist:** the modules *focus attention* on challenging regions, e.g. the corners of the arena and the surface of the fixed ball.

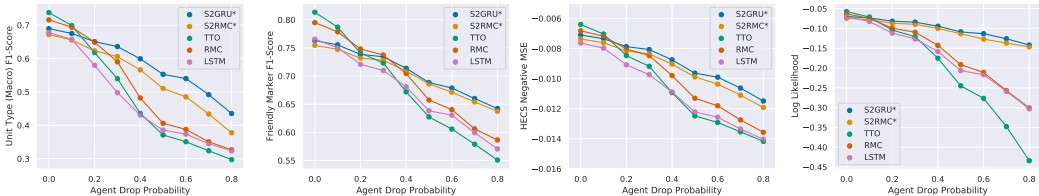

Figure 6: Ablation over the local and the non-local terms in the input and inter-cell attention mechanisms (KMDPAs). For a set number of bouncing balls, each sub-plot shows how the balanced accuracy changes with the fraction of views (crops) available to the model. **Gist:** Both local and non-local terms in KMDPA contribute to the overall performance. The non-local term is more important for the input attention, whereas the local term is more important for the inter-cell attention.

Figure 8: Performance metrics (larger the better) as a function of the probability that an agent will not supply information to the world model but still query it. **Gist:** while all models lose performance as fewer agents share observations, we find S2RMs to be most robust.

all non-oracle baselines on the one-step forward prediction task and strike a good balance with regard to in-distribution and OOD performance. In Figure 3, we qualitatively show reconstructions from 25 step rollouts on the out of distribution dataset with 5 balls to demonstrate that S2GRUs can perform OOD rollouts over long horizons without losing track of balls.

Figure 6 shows the result of an ablation study where we disable the local and non-local terms in each of the two KMDPAs while keeping everything else same. We see that both local and non-local terms contribute to the overall performance; moreover, the input attention relies on the non-local term and the performance severely affected by its absence, whereas the inter-cell attention is dependent on the local-term to yield good performance. This suggests that the modules indeed rely on the content of the input observations as they select their inputs, and learning an interaction topology between modules is a strong contributor to the final performance. In Figure 5, we show for each module its corresponding enclave, which is the spatial region that it is *responsible* for modelling, i.e. for pixels at position $\mathbf{x}$, we plot $\{Z(P(\mathbf{x}), \mathbf{p}^m)\}_{m=1}^{10}$ (cf. Section 4). We find that the modules learn to *share the responsibility* of modelling the entire spatial domain. Finally, in Figure 7 we see the effect of removing (randomly sampled) modules at test time, i.e. without additional retraining. The performance degrades gracefully as fewer modules are available, suggesting that the individual modules can function while other modules are missing. We include details and additional results in Appendix F.1.

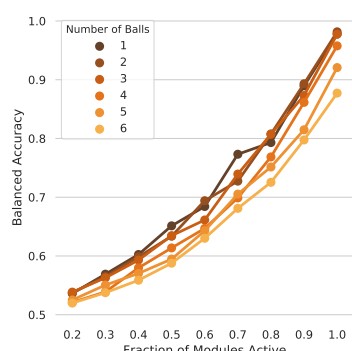

Figure 7: The effect of removing random modules at test time. **Gist:** Performance degrades gracefully as modules are removed, suggesting that modules can function even when their counterparts are removed, and that there is limited co-adaptation between them.

**Multi-Agent World Modeling on Starcraft2.** In Section 2, we formulated the problem of modeling what we called the *world state* $\mathfrak{o}$ of a dynamical system $\phi$ given *local observations* $\{(\mathbf{x}_t^a, \mathbf{O}_t^a)\}_{a=1}^A$ where $\mathbf{O}_t^a = \phi(t, \mathfrak{o})(\mathbf{x}_t^a)$. Under certain restrictions, this problem can be mapped to that of multi-agent world modeling from partial and local observations, allowing us to evaluate the proposed model in a rich and challenging setting. In particular, we consider environments that are **(a)** *spatial*, i.e. all agents $a$ have a well-defined and known location $\mathbf{x}_t^a$ (at time $t$), **(b)** the agents' actions $\mathbf{u}_t^a$ are local, in that their effects propagate away (from the agent) only at a finite speed, **(c)** the observations are local and centered around agents, in the sense that the agent only observes the events in its local vicinity, i.e., $\mathbf{O}_t^a$. Observe that we do not fix the number of agents in the environment, and allow

for agents to dynamically enter or exit the environment. Now, the task is: given observations $\mathbf{O}_t^a$ from a team of (cooperating) agents at position $\mathbf{x}_t^a$ and their corresponding actions $\mathbf{u}_t^a$, predict the observation $\mathbf{O}_{t'}^q$ that would be made by an agent at time $t' = t + 1$ if it were at position $\mathbf{x}^q$.

Starcraft2 unit-micromanagement (Samvelyan et al., 2019) is a multi-agent reinforcement learning benchmark, wherein teams of heterogeneously typed units must defeat a team of opponents in melee and ranged combat.

The observations $\mathbf{O}_t^a$ and actions $\mathbf{u}_t^a$ are both multi-channel images represented in polar coordinates centered around the agent position $\mathbf{x}_a^t$. The field of view (FOV) of each agent is therefore a circle of fixed radius centered around it. The channels of the image correspond to **(a)** a binary indicator marking whether a position in FOV is occupied by a living friendly agent (*friendly marker*), **(b)** a categorical indicator marking the type of living units at a given position in FOV (*unit-type marker*), and **(c)** four channels marking the health, energy, weapon-cooldown and shields (*HECS markers*) of all agents in FOV. With a heuristic, we gather a total of 9K trajectories $(\{\mathbf{x}_t^a, \mathbf{O}_t^a, \mathbf{u}_t^a\}_{a=1}^A)_{t=1}^{100}$ spread over three training *scenarios*, corresponding to `1c3s5z`[3], `3s5z` and `2s5z` in Samvelyan et al. (2019). We also sample 1K trajectories (each) from two OOD scenarios `1s2z` and `5s3z`. Details in Appendix B.1.

|  | UT-F1 | FM-F1 | NMSE | LL |
|---|---|---|---|---|
| (1s2z) | | | | |
| LSTM | 0.6267 | 0.8464 | -0.0040 | -0.0382 |
| RMC | 0.6839 | 0.8597 | -0.0033 | -0.0334 |
| S2GRU | **0.7488** | **0.8627** | **-0.0023** | **-0.0233** |
| S2RMC | 0.7317 | 0.8563 | -0.0026 | -0.0261 |
| (TTO) | 0.7518 | 0.8883 | -0.0025 | -0.0259 |
| (5s3z) | | | | |
| LSTM | 0.4975 | 0.7123 | -0.0134 | -0.1251 |
| RMC | **0.5414** | **0.7486** | -0.0132 | -0.1167 |
| S2GRU | 0.5310 | 0.7058 | **-0.0119** | **-0.1108** |
| S2RMC | 0.5114 | 0.6945 | -0.0124 | -0.1205 |
| (TTO) | 0.6115 | 0.7872 | -0.0107 | -0.0940 |

Table 1: Performance metrics on OOD scenarios `1s2z` and `5s3z` (larger numbers are better): unit-type macro F1 score (UT-F1), friendly-marker F1 score (FM-F1), HECS Negative Mean Squared Error (NMSE) and Log Likelihood (LL).

Having trained all models on scenarios `1c3s5z`, `3s5z` and `2s5z`, we test their robustness to dropped agents (Figure 8) and their performance on OOD scenarios (Table 1). We only include baselines that achieve similar or better validation scores than S2RMs. Figure 8 shows that S2RMCs remain robust when fewer agents supply their observations to the world model, whereas Table 1 shows that S2GRUs outperforms the baselines in the OOD scenario `1s2z` but is matched by RMCs in `5s3z` (see Appendix F.2 for details). The strong performance of RMCs suggests that the task benefits from the inductive bias of relational memory. One hypothesis as to why is that the pace of the considered environments requires fast communication between agents, which can be achieved by a shared memory where all agents may read from and write to. Further, we observe that while the oracle (TTO) can generalize well out of distribution, Figure 8 shows that it is less robust to the number of available observations. This is explained by the fact that unlike recurrent models, TTO does not leverage the temporal dynamics to fill in the missing information due to fewer available observations. This pattern also holds for the bouncing balls task, cf. Figures 21e and 20e in Appendix F.1.

## CONCLUSIONS, LIMITATIONS AND FUTURE WORK

We proposed Spatially Structured Recurrent Modules, a new class of models constructed to jointly leverage both spatial and modular structure in data, and explored its potential in the challenging problem setting of predicting the forward dynamics from partial observations at known spatial locations. In the tasks of video prediction from crops and multi-agent world modeling in the Starcraft2 domain, we found that it compares favorably against strong baselines in terms of out-of-distribution generalization and robustness to the number of available observations. Future work may attempt to extend the idea to parallel-in-time methods like universal transformers (Dehghani et al., 2018) and thereby address the computational bottleneck of recurrent processing, which is a current limitation. Another interesting avenue of research could be to explore how latent random variables can be used in tandem with the spatial structure to obtain a variational version of S2RMs. Finally, efficient implementations using block-sparse methods (Gray et al., 2017) might hold the key to unlock applications to significantly larger scale spatiotemporal forecasting problems encountered in domains like climate change research (Rolnick et al., 2019).

---

[3]Here, the code `1c3s5z` refers to a scenario where each team comprises 1 *colossus* (`1c`), 3 *stalkers* (`3s`), and 5 *zealots* (`5z`).

## ACKNOWLEDGEMENTS

The authors would like to thank Georgios Arvanitidis, Luigi Gresele, Michael Cobos for their feedback on the paper, and Murray Shanahan for the discussions. The authors also acknowledge the important role played by their colleagues at the Empirical Inference Department of MPI-IS Tübingen and Mila throughout the duration of this work.

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

## A  Qualitative Visualizations a Grid-World Navigation Task

In this section, we show qualitative results on a grid-world task defined in Eichenberger et al. (2019), which formulates the problem of navigation on a railway network in a multi-agent reinforcement learning framework. The environment comprises a network of railroads, on which agents (trains) may move in order to reach their destination. In our experiments, the entire railway network is defined on a $60 \times 60$ grid-world and we let each agent only observe a partial and local view of the environment, which is a $5 \times 5$ crop centered around itself.

We gather 10000 multi-agent trajectories with 10 agents and maximum length 128, from which we use 8000 for training and reserve 2000 for validation. We train S2GRU with 10 modules for 100 epochs and early stop when the validation loss is at its minimum. With the trained model, we visualize the following two things.

First: for each module $F_m$, we visualize the spatial locations it may attend to. To this end, we consider all $60 \times 60 = 3600$ pixel locations in the grid-world, say $\mathbf{x}_{ij}$ where $i, j \in \{1, ..., 60\}$. For each such $\mathbf{x}_{ij}$, we evaluate the quantity:

$$\mathcal{X}_{ij}^m = Z(\mathbf{p}^m, \mathbf{s}_{ij}) \tag{11}$$

where $\mathbf{s}_{ij} = P(\mathbf{x}_{ij})$ (see Eqn 3), $\mathbf{p}^m$ is the embedding of module $F_m$ and $\mathcal{X}^m$ is a $60 \times 60$ image indexed by $i$ and $j$, which we call the *enclave* of module $F_m$. Note that this is identical to what we visualize in Figure 5.

Next: we identify each module with its enclave, and visualize the graph of interactions between them. In Figure 9, we plot as nodes the enclaves $\mathcal{X}^m$. Further, we draw an edge between enclaves $\mathcal{X}^m$ and $\mathcal{X}^n$ iff $Z(\mathbf{p}^m, \mathbf{p}^n) > 0$.

We make the following two observations. First, the images in Figure 9 show that each module learns to account for a spatial region in the environment, as we imagined in Figure 1b in Section 1. Second, we find that the modules interact sparsely with each other – while some modules learn to interact with up-to five other modules, other modules learn to operate independently.

## B  Detailed Task Descriptions

### B.1  Starcraft2

The Starcraft2 Environment we use is a modified version of the SMAC-Env proposed in Samvelyan et al. (2019) and built on PySC2 wrapper around Blizzard SC2 API (Vinyals et al., 2017). Starcraft2 is a real-time-strategy (RTS) game where players are tasked with manufacturing and controlling armies of *units* (airborne or land-based) to defeat the opponent's army (where the opponent can be an AI or another human). The players must choose their *alien race*[4] before starting the game; available options are *Protoss*, *Terran* and *Zerg*. All unit types (of all races) have their strengths and weaknesses against other unit types, be it in terms of maximum health, shields (Protoss), energy (Terran), DPS (damage per second, related to weapon cooldown), splash damage, or manufacturing costs (measured in *minerals* and *vespene gas*, which must be mined).

The key engineering contribution of Samvelyan et al. (2019) is to repurpose the RTS game as a multi-agent environment, where the individual units in the army become individual agents[5]. The result is a rich and challenging environment where heterogeneous teams of agents must defeat each other in melee and ranged combat. The composition of teams vary between *scenarios*, of which Samvelyan et al. (2019) provide a selection. Further, new scenarios can be easily created with the SC2MapEditor, which allows for practically endlessly many possibilities.

We build on Samvelyan et al. (2019) by modifying their environment to better expose the transfer and out-of-distribution aspects of the domain by (a) standardizing the state and action space across a large class of scenarios and (b) standardizing the unit stats to better reflect the game-defined notion of hit-points.

---

[4]Please note that this is a game-specific notion.

[5]Note that this is rather unconventional, since each player usually controls entire armies and must switch between macro- and micro-management of units or unit-groups.

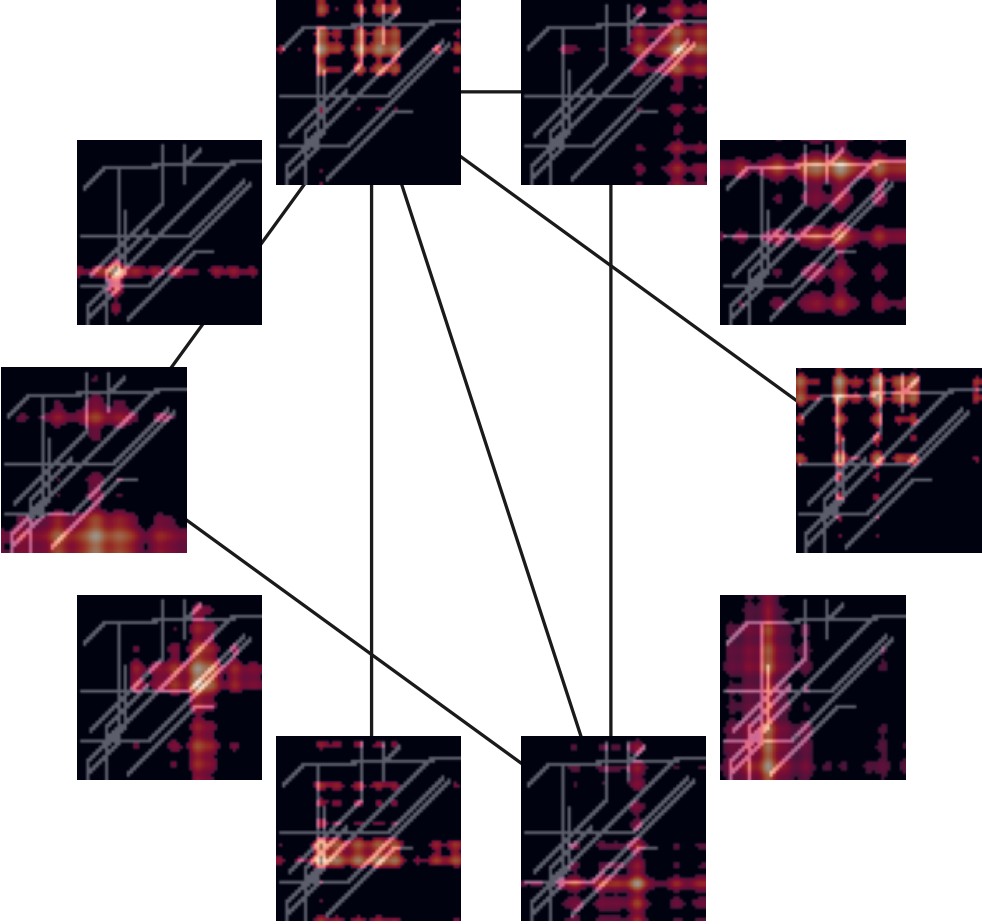

Figure 9: Joint visualization of spatial *enclaves* and the interaction graph between modules in the grid-world environment of Eichenberger et al. (2019), as detailed in Appendix A. The images show which spatial locations a module attends to via the local attention (spatial enclaves), whereas the presence of an edge indicates that the corresponding modules may interact via inter-cell attention. **Gist:** The modules indeed learn a notion of spatial locality, while interacting sparsely with each other.

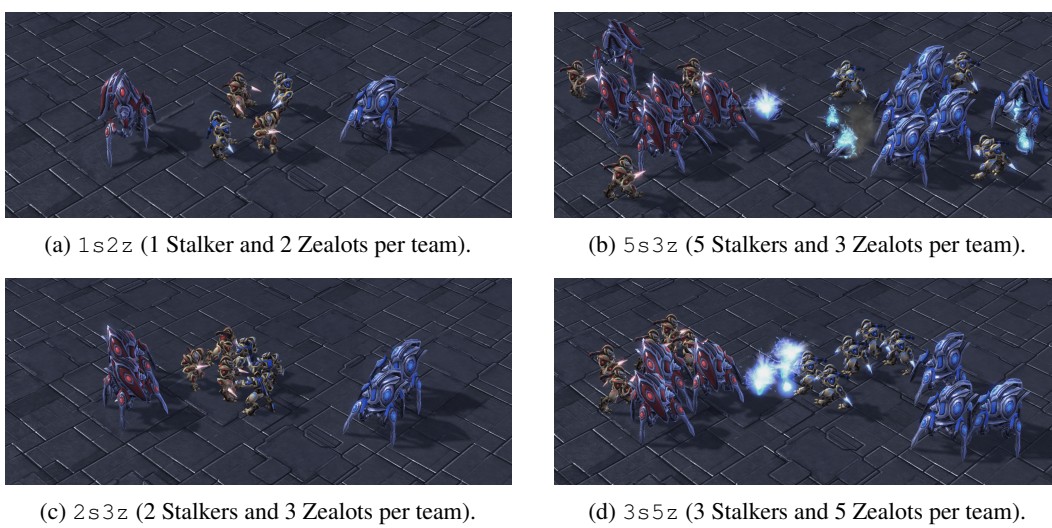

(a) `1s2z` (1 Stalker and 2 Zealots per team).

(b) `5s3z` (5 Stalkers and 3 Zealots per team).

(c) `2s3z` (2 Stalkers and 3 Zealots per team).

(d) `3s5z` (3 Stalkers and 5 Zealots per team).

(e) `1c3s5z` (1 Colossus, 3 Stalkers and 5 Zealots per team).

Figure 10: Human readable illustrations of the Starcraft2 (SMAC) scenarios we consider in this work. Figures 10a and 10b show the OOD scenarios, whereas Figures 10c, 10d and 10e show the training scenarios (provided by Samvelyan et al. (2019)).

### B.1.1 STANDARDIZED STATE SPACE FOR ALL SCENARIOS

In the environment provided by Samvelyan et al. (2019), the dimensionality of the vector state space varies with the number of friendly and enemy agents, which in turn varies with the scenario. While this is not an issue in the typical use case of training MARL agents in a fixed scenario, it is not convenient for designing models that seamlessly handle multiple scenarios. In the following, we propose an alternate state representation that preserves the spatial structure and is consistent across multiple scenarios.

Instead of representing the state of an agent $a$ with a vector of variable dimension, we represent it with a multi-channel *polar image* $\mathfrak{I}^a$ of shape $C \times I \times J$, where $C$ is the number of channels and $(I, J)$ is the image size. Given the *radial* and *angular* resolutions $\rho$ and $\varphi$ (respectively), the pixel coordinate $i = 0, ..., I - 1, j = 0, ..., J - 1$ corresponds to coordinates $(i \cdot \rho, j \cdot \varphi)$ with respect to a polar coordinate system centered on the agent $a$, where the positive $x$-axis ($j = 0$) points towards the east. Further, the field of view (FOV) of an agent is characterized by a circle of radius $I \cdot \rho$ centered on the agent at 2D game-coordinates $\mathbf{x}^a = (x_1^a, x_2^a)$, to which the Starcraft2 API (Vinyals et al., 2017) provides raw access.

The polar image $\mathfrak{I}^a$ therefore provides an agent-centric view of the environment, where pixel coordinates $i, j$ in $\mathfrak{I}^a$ can be mapped to global game coordinates $\mathbf{x} = (x_1, x_2)$ in FOV via:

$$x_1 = i \cdot \rho \cos\left[j \cdot \varphi\right] + x_1^a \tag{12}$$
$$x_2 = i \cdot \rho \sin\left[j \cdot \varphi\right] + x_2^a \tag{13}$$

In what follows, we denote this transformation with $T_a$, as in $T_a(i, j) = (x_1, x_2)$.

Now, the channels in the polar image can encode various aspects of the observation; in our case: friendly markers (one channel), unit-type markers (nine channels, one-hot), health-energy-cooldown-

shields (HECS, four channels) and terrain height (one channel). As an example, let us consider the friendly markers, which is a binary indicator marking units that are friendly. If we have an agent at game position $(x_1, x_2)$ that is friendly to agent $a$, then we would expect the pixel coordinate $(i, j) = T_a^{-1}(x_1, x_2)$ of the corresponding channel in the polar image $\mathfrak{I}^a$ to be 1, but 0 otherwise. Likewise, the value of $\mathfrak{I}$ at the channels corresponding to HECS at pixel position $i, j$ gives the HECS of the corresponding unit[6] at $T_a(i, j)$. This representation has the following advantages: **(a)** it does not depend on the number of units in the field of view, **(b)** it exposes the spatial structure in the arrangement of units which can naturally processed by convolutional neural networks (e.g. with circular convolutions).

Nevertheless, it has the disadvantage that the positions are *quantized* to pixels, but the euclidean distance between the locations represented by pixels $(i, j)$ and $(i, j + 1)$ increases with increasing $i$. Consequently, this representation may not remain suitable for larger FOVs.

Further, this representation is also appropriate for the action space. Given an agent, we represent the one-hot categorical actions of all friendly agents in FOV as a multi-channel polar image. In this representation, the pixel position $i, j$ gives the action taken by an agent at at position $T_a(i, j)$. Unfriendly agents get assigned an "unknown action", whereas positions not occupied by a living agent are assigned a "no-op" action.

### B.1.2 STANDARDIZED UNIT STATS

At any given point in time, an active unit in Starcraft2 has certain *stats*, e.g. its health, energy (Terran), shields (Protoss) and weapon-cooldown (for armed units). A large and expensive unit-type like the Colossus has more max-health (*hit-points*) than smaller units like Stalkers and Marines[7]. Likewise, unit-types differ in the rate at which they deal damage (measured in damage-per-second or DPS, excluding splash damage), which in turn depends on the cooldown duration of the active weapon.

Now, the environment provided by Samvelyan et al. (2019) normalizes the stats by their respective maximum value, resulting in values between 0 and 1. However, given that different units may have different normalization, the stats are rendered incomparable between unit types (without additionally accounting the unit-type). We address this by standardizing stats (instead of normalizing) by dividing them by a fixed value. In this scheme, the stats are scaled uniformly across all unit-types, enabling models to directly rely on them instead of having to account for the respective unit-types.

### B.2 VIDEO PREDICTION FROM CROPS ON THE BOUNCING BALLS TASK

The bouncing balls task is a well-known test-bed for evaluating the performance of video prediction models (Fraccaro et al., 2017; Watters et al., 2017; Miladinović et al., 2019; Kossen et al., 2019; Cenzato et al., 2019). We modify the problem by introducing partial observability – concretely, instead of providing the model with the full image frames, we only provide it with crops at randomly sampled locations.

As mentioned in Section 6, at every time step $t$ we sample $A = 10$ pixel positions $\{\mathbf{x}_t^a\}_{a=1}^{10}$ from the $t$-th full video frame $\mathfrak{o}_t$ of size $48 \times 48$. Around the sampled central pixel positions $\mathbf{x}_t^a$, we extract $11 \times 11$ crops, which we use as the *local* observations $\mathbf{O}_t^a$. The task now is to predict $11 \times 11$ crops $\mathbf{O}_{t'}^q$ corresponding to query central pixel positions $\mathbf{x}_{t'}^q$ at a future time-step $t' > t$. Observe that at any given time-step $t$, the model has access to at most 52% of the global video frame assuming that the crops never overlap (which is rather unlikely).

We train all models on a training dataset of 20K video sequences with 100 frames of 3 balls bouncing in an arena of size $48 \times 48$. We also include an additional fixed ball in the center to make the task more challenging. We use another 1K video sequences of the same length and the same number of balls as a held-out validation set. In addition, we also have 5 out-of-distribution (OOD) test sets with various number of bouncing balls (ranging from 1 to 6) and each containing 1K sequences of length 100.

---

[6]If health drops to zero, the unit is considered dead and the representation does not differentiate between dead and absent units.

[7]These stats may change with game-versions, and are catalogued here: `https://liquipedia.net/starcraft2/Units_(StarCraft)`.

In Figure 4, for each number of balls (i.e. point on the x-axis), we plot the respective metrics which are aggregated over 10 randomly selected $11 \times 11$ crops of a total of 100000 frames spread over 1000 trajectories with 100 frames each.

## C    PRECISE DESCRIPTION OF ATTENTION MECHANISMS

### C.1    INPUT ATTENTION

Recall from Section 4 that the input attention mechanism is a mapping between sets: namely, from that of observation encodings $\{\mathbf{e}_t^a\}_{a=1}^A$ to that of RNN inputs $\{\mathbf{u}_t^m\}_{m=1}^M$. In what follows, we use the einsum notation[8] to succinctly describe the exact mechanism. But before that, we repeat the definition of the truncated spherical Gaussian kernel (Fasshauer, 2011) to quantify the similarity between two points $\mathbf{p}, \mathbf{s} \in \mathcal{S}$:

$$Z(\mathbf{p}, \mathbf{s}) = \begin{cases} \exp\left[-2\epsilon(1 - \mathbf{p} \cdot \mathbf{s})\right], & \text{if } \mathbf{p} \cdot \mathbf{s} \geq \tau \\ 0, & \text{otherwise} \end{cases} \tag{14}$$

where $\epsilon \in \mathbb{R}^+$ and $\tau \in [-1, 1)$ are hyper-parameters (*kernel bandwidth* and *truncation parameter*, respectively), and $0 \leq Z \leq 1$ since $\mathbf{p}$ and $\mathbf{s}$ are unit vectors. We observe that both $\tau$ and $\epsilon$ controls the sparsity of the kernel: $\tau$ determines the size of the neighborhood of $\mathbf{p}$, i.e. the size of the set $\mathcal{B}(\mathbf{p}) \subset \mathcal{S}$ of all $\mathbf{s} \in \mathcal{B}(\mathbf{p})$ such that $\mathbf{p} \cdot \mathbf{s} \geq \tau$ and accordingly $Z(\mathbf{p}, \mathbf{s}) > 0$, whereas the bandwidth $\epsilon$ controls how the attention decays inside $\mathcal{B}(\mathbf{p})$. Intuitively, $\tau$ determines a lower bound to the amount of sparsity that the kernel induces (irrespective of the bandwidth $\epsilon$), whereas for fixed $\tau$, sparsity can be increased by increasing $\epsilon$. We find $\tau \in [-1, 0.6]$ and $\epsilon \in [0.9, 2]$ to work well; setting $\tau$ and $\epsilon$ to much larger values destabilizes the training due to excessive sparsity, whereas setting $\epsilon$ to much smaller values results in $Z$ being flat inside $\mathcal{B}(\mathbf{p})$ and therefore poor propagation of gradients.

Now, we use $k$ to index the attention heads, $d$ to index the dimension of the key and query vectors, and denote with $e_{ai}$ the $i$-th component of $\mathbf{e}_t^a$ and with $h_{mj}$ the $j$-th component of $\mathbf{h}_t^m$. Given learnable parameters $\Theta^{(K)}, \Theta^{(Q)}, \Theta^{(V)}$, we obtain:

$$Q_{akd} = e_{ai}\Theta_{ikd}^{(Q)} \qquad K_{mkd} = h_{mj}\Theta_{jkd}^{(K)} \tag{15}$$

$$V_{akv} = e_{ai}\Theta_{ikv}^{(V)} \qquad \tilde{W}_{mak} = Q_{akd}K_{mkd} \tag{16}$$

$$\bar{W}_{mak} = \text{sm}_a(\tilde{W}_{mak}) \qquad W_{ma}^{(L)} = Z(\mathbf{p}^m, \mathbf{s}^a) \tag{17}$$

$$W_{mak} = W_{ma}^{(L)}\bar{W}_{mak} \qquad \tilde{u}_{m(kv)} = W_{mak}V_{akv} \tag{18}$$

where: $\text{sm}_a$ denotes softmax along the $a$-dimension, $W^{(L)}$ is what we will call the *local weights*, we omit the time subscript in $\mathbf{s}^a$ for notational clarity, and $\tilde{u}_{m(kv)}$ is the $(kv)$-th component of a vector $\tilde{\mathbf{u}}^m$. Finally, we obtain the components $u_{mi}$ of RNN inputs $\mathbf{u}_t^m$ via a gating operation:

$$u_{mi} = G_m^{(\text{inp})} \cdot b_{mi} + (1 - G_m^{(\text{inp})}) \cdot \tilde{u}_{mi} \tag{19}$$

where the gating weight $G_m^{(\text{inp})} \in (0, 1)$ is obtained by passing $\tilde{u}_{mi}$ and $b_{mi} = W_{ma}^{(L)}e_{ai}$ through a two-layer MLP with sigmoidal output (in parallel across $m$). Now, observe that by weighting the MHDPA attention outputs ($\bar{W}$ in Equation 18) by the kernel $Z$ (via $W^{(L)}$), we construct a scheme where the interaction between input $\mathbf{O}_t^a$ and RNN $F_m$ is allowed only if the embedding $\mathbf{s}_t^a$ of the corresponding position $\mathbf{x}_t^a$ has a large enough cosine similarity ($\geq \tau$) to the embedding $\mathbf{p}_m$ of $F_m$. This partially implements the assumption of *Locality of Observation* detailed in Section 3.

### C.2    INTER-CELL ATTENTION

Recall from Section 4 that the inter-cell attention maps the hidden states of each RNN $\{\mathbf{h}_t^m\}_{m=1}^M$ to the set of *aggregated hidden states* $\{\bar{\mathbf{h}}_t^m\}_{m=1}^M$, thereby enabling interaction between the RNNs $F_m$. While its mechanism is identical to that of the input attention, we formulate it below for completeness.

---

[8]Indices not appearing on both sides of an equation are summed over; this is implemented as `einsum` in most DL frameworks.

To proceed, we denote with $h_{li}$ the $i$-th component of $\mathbf{h}_t^l$ (in addition to the notation introduced before Equation 15), and take $\Phi^{(Q)}$, $\Phi^{(K)}$ and $\Phi^{(V)}$ to be learnable parameters. We have:

$$Q_{mkd} = h_{mj}\Phi_{jkd}^{(Q)} \qquad K_{lkd} = h_{li}\Phi_{ikd}^{(K)} \tag{20}$$

$$V_{lkv} = h_{li}\Phi_{ikv}^{(V)} \qquad \tilde{W}_{mlk} = Q_{mkd}K_{lkd} \tag{21}$$

$$\bar{W}_{mlk} = \mathrm{sm}_l(\tilde{W}_{mlk}) \qquad W_{ml}^{(L)} = Z(\mathbf{p}^m, \mathbf{p}^l) \tag{22}$$

$$W_{mlk} = \bar{W}_{mlk}W_{ml}^{(L)} \qquad \tilde{h}_{m(kv)} = W_{mlk}V_{lkv} \tag{23}$$

where $\tilde{h}_{m(kv)}$ is the $(kv)$-th component of a vector $\tilde{\mathbf{h}}^m$. Finally, the $j$-th component $\bar{h}_{mj}$ of the aggregated hidden state $\bar{\mathbf{h}}_t^m$ in Equation 4 is given by a gating operation:

$$\bar{h}_{mj} = G_m^{(\mathrm{ic})} \cdot c_{mj} + (1 - G_m^{(\mathrm{ic})}) \cdot \tilde{h}_{mj} \tag{24}$$

where the gating weight $G_m^{(\mathrm{ic})} \in (0,1)$ is obtained by passing $\tilde{h}_{mj}$ and $c_{mj} = W_{ml}^{(L)}h_{lj}$ through a two-layer MLP with sigmoid output (in parallel across $m$). The weighting by $Z$ (in Equation 23, left) ensures that the interaction is constrained to be only between RNNs whose embeddings in $\mathcal{S}$ are similar enough, thereby implementing the assumption of *Local Interactions between Sub-systems* in Section 3.

## C.3 Positional Encoding

In Section 4, recall that we used the following positional embedding $P$:

$$P(\mathbf{x}) = {}^{\mathbf{s}}/_{\|\mathbf{s}\|} \in \mathcal{S} \quad \text{where} \quad (s_{i+m}, s_{i+1+m}) = (\sin\left(x_m/10000^i\right), \cos\left(x_m/10000^i\right)) \tag{25}$$

In this section, we explore how the choice of a positional embedding function $P$ determines a function space of spatial functions (defined on $\mathcal{X}$) that the local-attention can represent. To this end, consider the distance in $\mathcal{S}$ of a module with embedding $\mathbf{p}$ to an observation made at location $\mathbf{x}$ as a function of $\mathbf{x}$, given by

$$w^{(L)}(\mathbf{x}) = \mathbf{p} \cdot P(\mathbf{x}) \tag{26}$$

Here, the local weight of interaction between the module at $\mathbf{p}$ and an observation made at $\mathbf{x}$ is given by:

$$Z(\mathbf{p}, P(\mathbf{x})) = \begin{cases} \exp\left[-2\epsilon(1 - w^{(L)}(\mathbf{x}))\right], & \text{if } w^{(L)}(\mathbf{x}) \geq \tau \\ 0, & \text{otherwise} \end{cases} \tag{27}$$

In particular, observe that in order for two locations $\mathbf{x}$ and $\mathbf{y}$ to be *connected* by the module, we require from $w^{(L)}$ that it be flexible enough such that $w^{(L)}(\mathbf{x}) \geq \tau$ and $w^{(L)}(\mathbf{y}) \geq \tau$ for a chosen $\tau$. This flexibility stems from the fact that we implicitly express $w^{(L)}$ as a linear combination of sinusoidal basis functions with learned weights:

$$w^{(L)}(\mathbf{x}) = \sum_{j=0}^{J} [p_{2j}\cos\left(\omega_j \cdot \mathbf{x}\right) + p_{2j+1}\sin\left(\omega_j \cdot \mathbf{x}\right)] \tag{28}$$

Here, $p_{2j}$ and $p_{2j+1}$ are learnable parameters (as components of learnable vector $\mathbf{p}$ of dimension $2J$), and $\omega_j$ are frequency vectors. Now, if the dimension of the embedding vector $\mathbf{p}$ were to tend to infinity, we may have a growing number of frequencies $\omega_j$ to gradually recover the full Fourier basis of $L^2(\mathcal{X})$ (assuming $\mathcal{X}$ is Euclidean for simplicity). In the limit, $w^{(L)}(\mathbf{x})$ can be an arbitrary function lying on a unit sphere in $L^2(\mathcal{X})$ (i.e. $\int |w^{(L)}|^2 = 1$; recall that $\mathbf{p}$ is normed to unity). In other words, in a large dimensional embedding space, the system is afforded a large amount flexibility to learn any spatial structure or topology on $\mathcal{X}$ by connecting pairs of points $\mathbf{x}$ and $\mathbf{y}$ in $\mathcal{X}$ via a module.

For computational tractability, however, we require $\mathbf{p}$ to be finite-dimensional, implying that $\omega_j$ must be sampled. By using $P$ as defined in Equation 25, we essentially sample $\omega_j$ as coordinate (one-hot) vectors with $\log\|\omega_j\|$ sampled on a uniform grid. This sampling step is not only computationally favorable, but also justified in the theory of RKHS – Rahimi & Recht (2007) use Bochner's theorem to show that any proper distribution $p(\omega)$ (from which $\omega$ can be sampled) leads to a feature map,

the inner product of which in expectation over $p$ corresponds to a positive-definite kernel. The convergence to such a kernel is exponential in the number of samples (equivalent to the dimension of the embedding).

Further, we note that while the sampling constrains the function space in which $w^{(L)}$ can lie in, we find (empirically) that this can in fact have a regularizing effect. Nevertheless, this raises the question whether other choices of a basis function are viable. We speculate that a polynomial basis (e.g. feature maps of a degree $d$ polynomial kernel) might also be viable, but leave extensive exploration to future work.

## D   BASELINE ARCHITECTURE

As mentioned in Section 6: in order to ensure fair comparison between the baselines and our method, we describe a baseline architecture constructed to satisfy a few critically important desiderata that are naturally satisfied by S2RMs. Namely, (a) it must be parameterically agnostic to the number of available observations and (b) it must be invariant to the permutation of the observations. For this, we extend the framework Generative Query Networks (Eslami et al., 2018) by predicting the forward dynamics of an *aggregated representation*. While we invest effort in ensuring that the resulting class of models can perform at least as well as S2RMs on in-distribution (validation) data, we do not consider it a novel contribution of this work.

**Encoder.** At a given timestep $t$, the encoder $E$ jointly maps the embedding $\mathbf{s}_t^a \in \mathcal{S}$ of the position $\mathbf{x}_t^a \in \mathcal{X}$ and the corresponding observations $\mathbf{O}_t^a$ to encodings $\mathbf{e}_t^a$, which are then summed over $a$ to obtain an *aggregated representation*:

$$\mathbf{r}_t = \sum_{a=1}^{A} E(\mathbf{O}_t^a, \mathbf{s}_t^a) \tag{29}$$

The additive aggregation scheme we use is well known from prior work (Santoro et al., 2017; Eslami et al., 2018; Garnelo et al., 2018) and makes the model agnostic to $A$ and to permutations of $(\mathbf{x}_t^a, \mathbf{O}_t^a)$ over $a$. The encoder $E$ is a seven-layer CNN with residual layers, and the positional embedding $\mathbf{s}_t^a$ is injected after the second convolutional layer via concatenation with the feature tensor. The exact architectures can be found in Appendices E.1 and E.2.

**RNN.** The aggregated representation $\mathbf{r}_t$ is used as an input to a RNN model $F$ as following:

$$\mathbf{h}_{t+1}, \mathbf{c}_{t+1} = F(\mathbf{r}_t, \mathbf{h}_t, \mathbf{c}_t) \tag{30}$$

where $\mathbf{h}_t$ and $\mathbf{c}_t$ are hidden and memory states of the RNN $F$ respectively. We experiment with various RNN models, including LSTMs (Hochreiter & Schmidhuber, 1997), RMCs (Santoro et al., 2018) and Recurrent Independent Mechanisms (RIMs) (Goyal et al., 2019).

As a sanity check, we also show results with a *Time Travelling Oracle* (TTO), which has access to $r_{t+1}$ (but at time step $t$), and produces $\mathbf{h}_{t+1} = F_{TTO}(\mathbf{r}_{t+1})$ with a two layer MLP $F_{TTO}$. TTO therefore does not model the dynamics, but merely verifies that the additive aggregation scheme (Equation 29) and the querying mechanism (Equation 31) are sufficient for the task at hand.

**Decoder.** Given the embedding $\mathbf{s}^q$ of the query position $\mathbf{x}^q$, the decoder $D$ predicts the corresponding observation $\hat{\mathbf{O}}_{t+1}^q$:

$$\hat{\mathbf{O}}_{t+1}^q = D(\mathbf{h}_{t+1}, \mathbf{s}^q) \tag{31}$$

We parameterize $D$ with a deconvolutional network with residual layers, and inject the positional embedding of the query $\mathbf{s}^q$ after a single convolutional layer by concatenating with the layer features (see Appendices E.1 and E.2).

## E   HYPERPARAMETERS AND ARCHITECTURES

### E.1   ENCODER AND DECODER FOR BOUNCING BALLS

The architectures of image encoder and decoder was fixed for all models after initial experimentation. We converged to the following architectures.

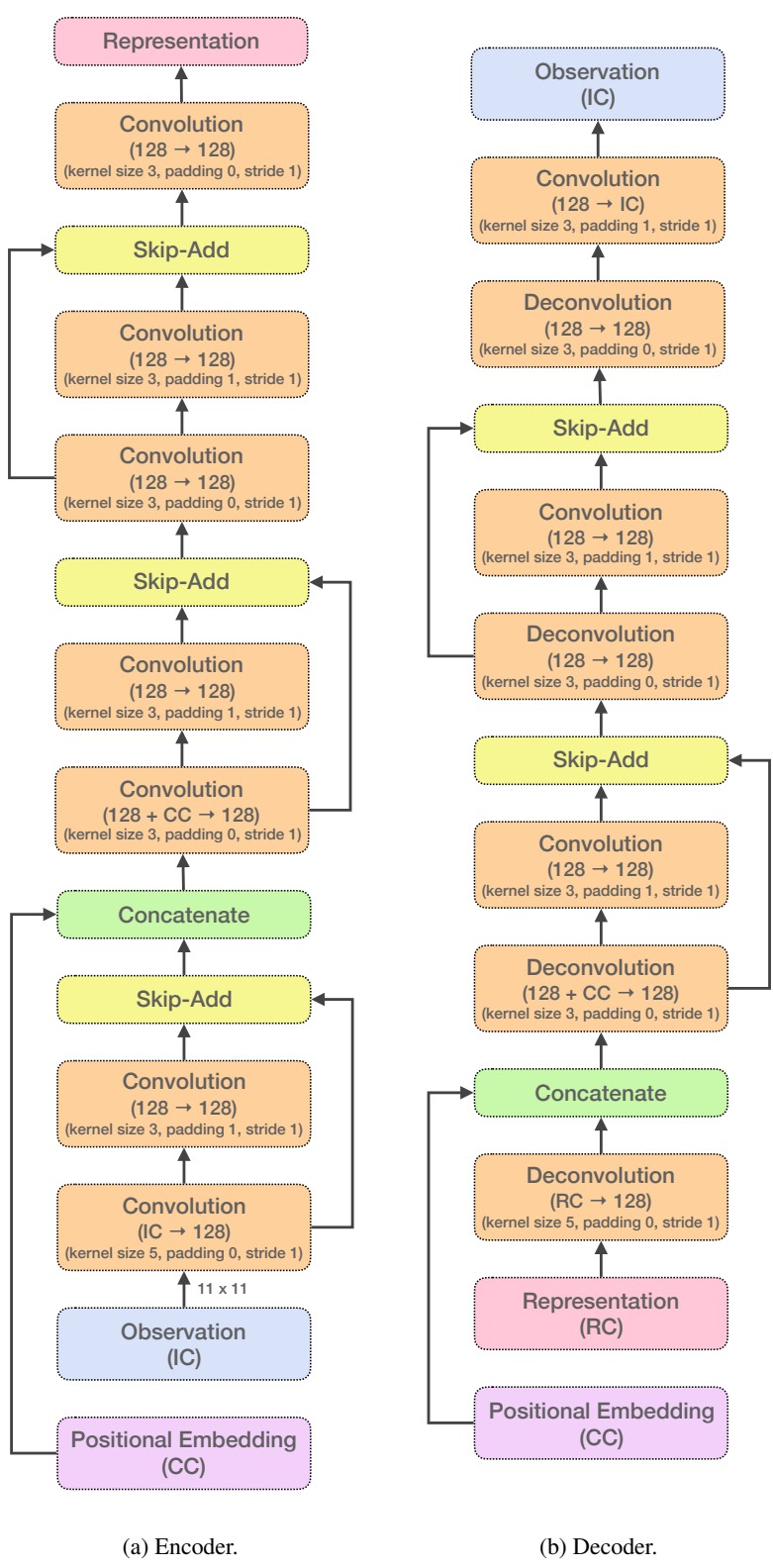

(a) Encoder.

(b) Decoder.

Figure 11: Baseline encoder and decoder architectures for the Bouncing Ball task.

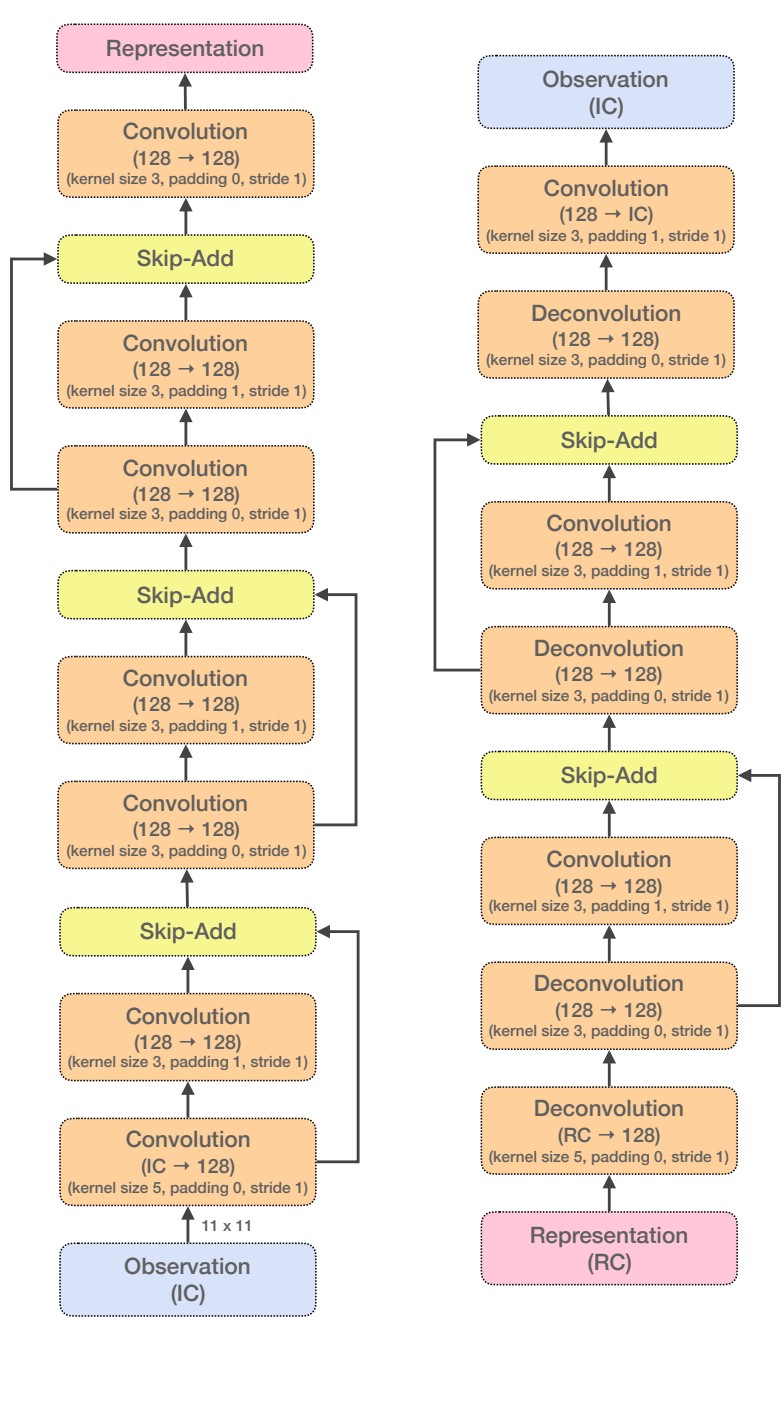

(a) Encoder.                    (b) Decoder.

Figure 12: S2RM encoder and decoder architectures for the Bouncing Ball task.

### E.1.1 S2RMs

The encoder (decoder) is a (de)convolutional network with residual connections (Figure 12).

### E.1.2 BASELINES

Like in the case of S2RMs, the encoder (decoder) is a (de)convolutional network with residual connections (Figure 11), but with the positional embeddings injected after the second convolutional layer. This is loosely inspired by the encoders used in Eslami et al. (2018).

## E.2 ENCODER AND DECODER FOR STARCRAFT2

### E.2.1 S2RMs

Recall from Appendix B.1 that the states are polar images. We therefore use *polar convolutions*, which entails zero-padding the input image along the first (*radial*) dimension but circular padding along the second (*angular*) dimension. The encoder and decoder architectures can be found in Figure 14.

### E.2.2 BASELINES

Like for S2RMs, we use polar convolutions while injecting the positional embeddings further downstream in the network. The corresponding encoder and decoder architectures are illustrated in Figure 13.

## E.3 SPATIALLY STRUCTURED RELATIONAL MEMORY CORES (S2RMCs)

Embedding Relational Memory Cores (Santoro et al., 2018) naïvely in the S2RM architecture did not result in a working model. We therefore had to adapt it by first projecting the memory matrix ($M$ in Santoro et al. (2018)) of the $m$-th RMC to a *message* $\mathbf{h}_t^m$. This message is then processed by the intercell attention to obtain $\bar{\mathbf{h}}_t^m$, which is finally concatenated with the memory matrix and current input before applying the attention mechanism (i.e. in Equation 2 of Santoro et al. (2018), we replace $[M; x]$ with $\left[M; x, \bar{\mathbf{h}}_t^m\right]$).

## E.4 HYPERPARAMETERS

### E.4.1 BOUNCING BALL MODELS

The hyperparameters we used can be found in Table 2. Further, note that in Equation 5, we pass the gradients through the constant region of the kernel as if the kernel had not been truncated.

### E.4.2 STARCRAFT2 MODELS

The hyperparameters we used can be found in Table 3. Note that we only report models that attained a validation loss similar to or better than S2RMs.

### E.4.3 TRAINING

All models were trained using Adam Kingma & Ba (2014) with an initial learning rate $0.0003$[9]. We use Pytorch's (Paszke et al., 2019) `ReduceLROnPlateau` learning rate scheduler to decay the learning rate by a factor of 2 if the validation loss does not improve by at least $0.01\%$ over the span of 5 epochs. We initially train all models for 100 epochs, select the best of three successful runs, fine-tune it for another 100 epochs, and finally select the checkpoint with the lowest validation loss (i.e. we early stop). We train all models with batch-size 8 (Starcraft2) or 32 (Bouncing Balls) on a single V100-32GB GPU (each).

---

[9]`https://twitter.com/karpathy/status/8016217641449971776?s=20`

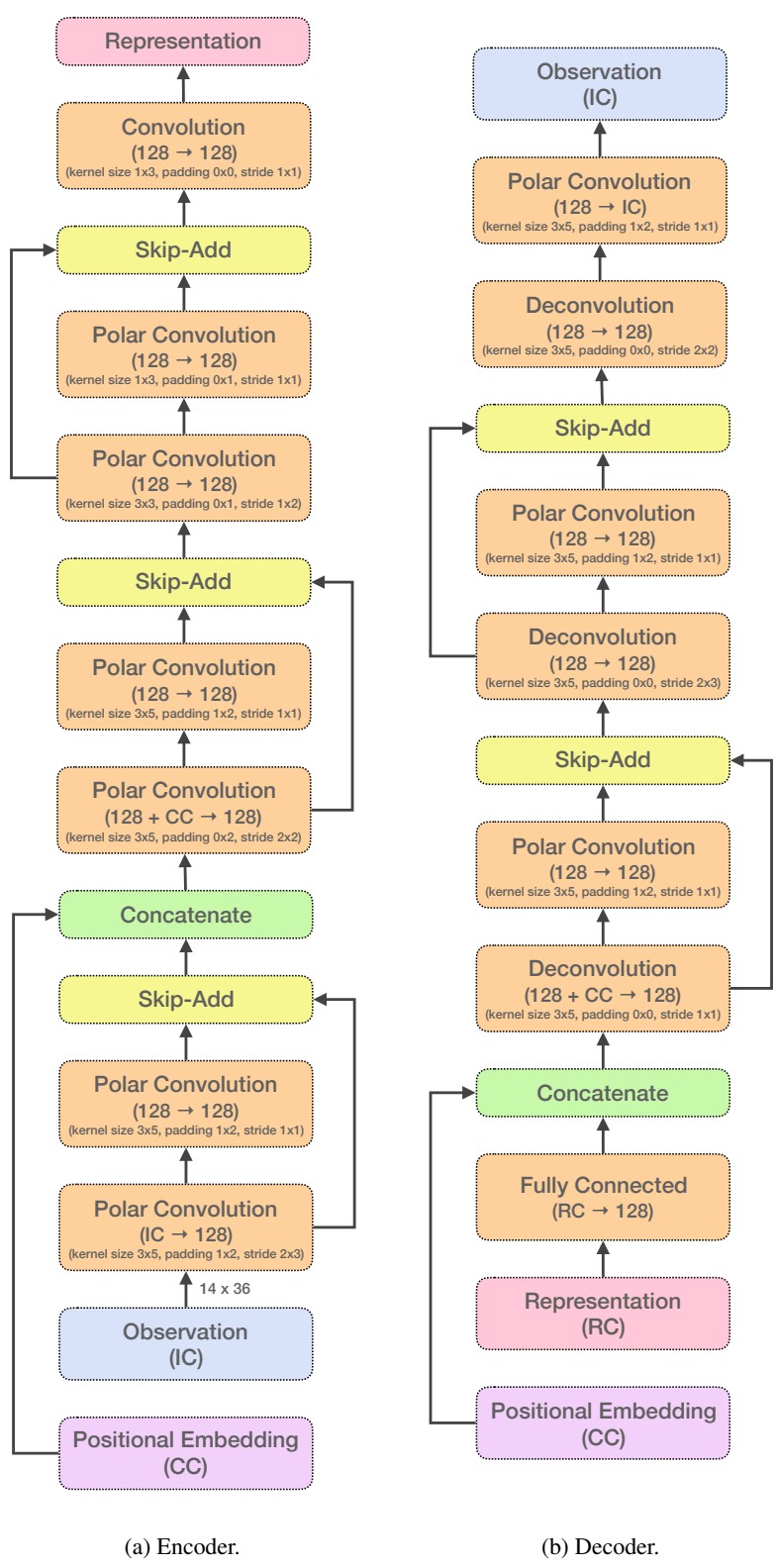

(a) Encoder.                    (b) Decoder.

Figure 13: Baseline encoder and decoder architectures for the Starcraft2 task.

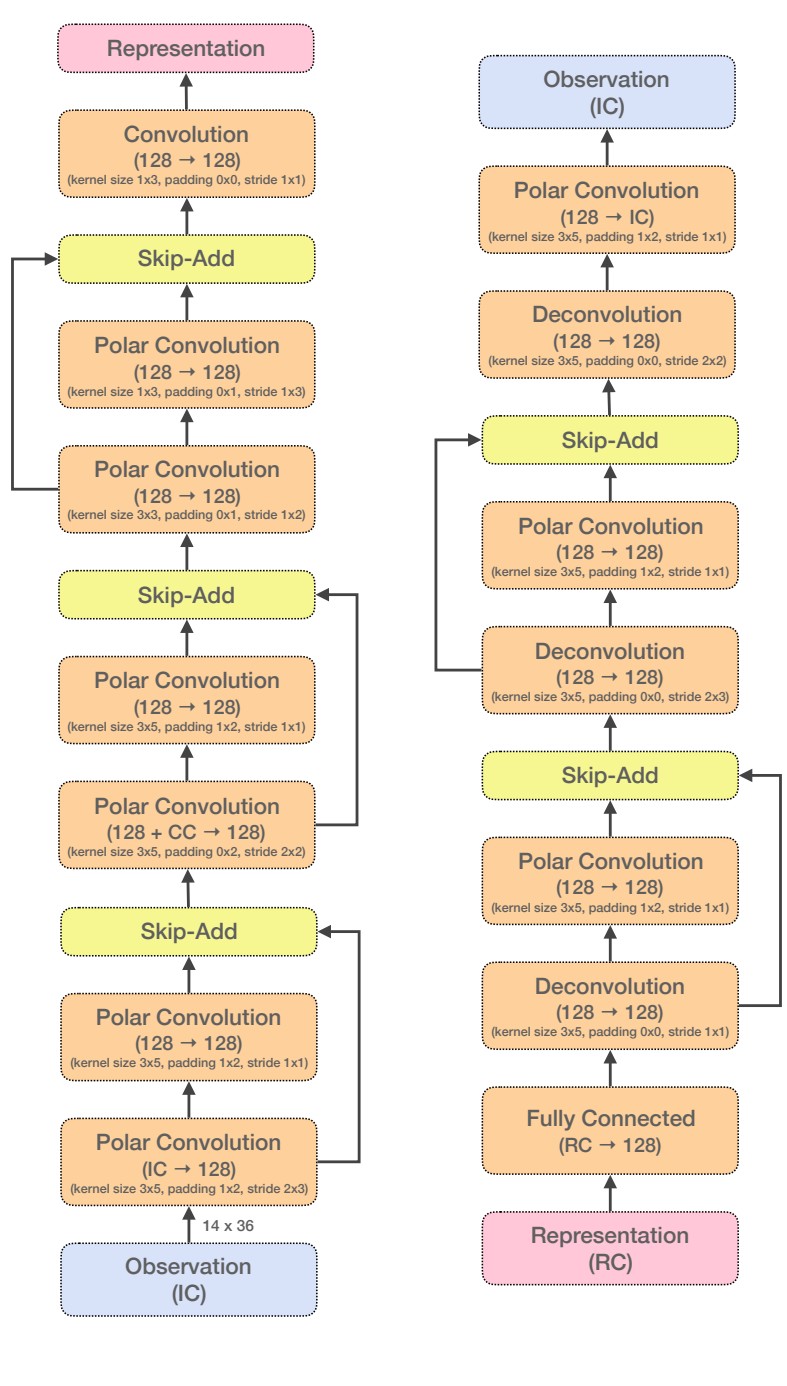

(a) Encoder.                              (b) Decoder.

Figure 14: S2RM encoder and decoder architectures for the Starcraft2 task.

| Model | |
|---|---|
| Hyperparameter | Value |
| **S2GRU** | |
| Number of modules ($M$) | 10 |
| GRU: hidden size per module | 128 |
| Module embedding size ($d$) | 16 |
| Kernel bandwidth ($\epsilon$) | 1 |
| Kernel truncation ($\tau$) | 0.6 |
| `shape` $\Theta^{(Q/K)}$ | (128, 2, 016) |
| `shape` $\Theta^{(V)}$ | (128, 2, 128) |
| `shape` $\Phi^{(Q/K)}$ | (128, 4, 016) |
| `shape` $\Phi^{(V)}$ | (128, 4, 128) |
| **RMC** (Santoro et al., 2018) | |
| Number of attention heads | 4 |
| Size of attention head | 128 |
| Number of memory slots | 1 |
| Key size | 128 |
| **LSTM** (Hochreiter & Schmidhuber, 1997) | |
| Hidden size | 512 |
| **RIMs** (Goyal et al., 2019) | |
| Number of RIMs ($k_T$) | 6 |
| Update Top-k ($k_A$) | 5 |
| Hidden size ($h_{size}$) | 510 |
| Input key size | 32 |
| Input value size | 400 |
| **TTO** | |
| MLP hidden size | 512 |

Table 2: Hyperparameters used for various models on the Bouncing Ball task. Hyperparameters not listed here were left at their respective default values.

| **Model** | |
|---|---|
| Hyperparameter | Value |
| **S2GRUs** | |
| Number of modules ($M$) | 10 |
| GRU: hidden size per module | 128 |
| Module embedding size ($d$) | 8 |
| Kernel bandwidth ($\epsilon$) | 1 |
| Kernel truncation ($\tau$) | 0.5 |
| `shape` $\Theta^{(Q/K)}$ | (128, 2, 016) |
| `shape` $\Theta^{(V)}$ | (128, 2, 128) |
| `shape` $\Phi^{(Q/K)}$ | (128, 4, 016) |
| `shape` $\Phi^{(V)}$ | (128, 4, 128) |
| **S2RMC** | |
| Number of modules ($M$) | 10 |
| RMC: number of attention heads | 4 |
| RMC: size of attention head | 64 |
| RMC: number of memory slots | 4 |
| RMC: key size | 64 |
| Module embedding size ($d$) | 8 |
| Kernel bandwidth ($\epsilon$) | 1 |
| Kernel truncation ($\tau$) | 0.5 |
| `shape` $\Theta^{(Q/K)}$ | (128, 2, 016) |
| `shape` $\Theta^{(V)}$ | (128, 2, 128) |
| `shape` $\Phi^{(Q/K)}$ | (128, 4, 016) |
| `shape` $\Phi^{(V)}$ | (128, 4, 128) |
| **RMC** (Santoro et al., 2018) | |
| Number of attention heads | 4 |
| Size of attention head | 128 |
| Number of memory slots | 1 |
| Key size | 16 |
| **LSTM** (Hochreiter & Schmidhuber, 1997) | |
| Hidden size | 2048 |
| **TTO** | |
| MLP hidden size | 512 |

Table 3: Hyperparameters used for various models on the Starcraft2 task. Hyperparameters not listed here were left at their respective default values.

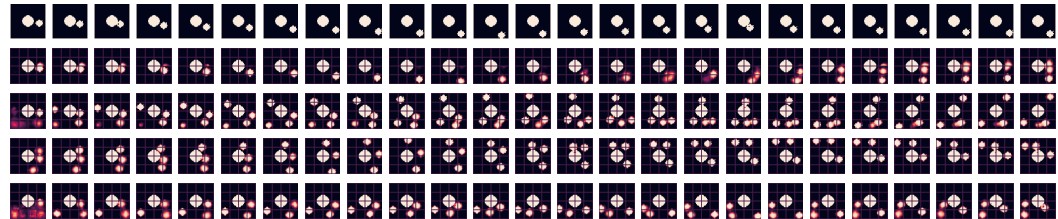

Figure 15: Rollouts (OOD) with 1 bouncing ball, from top to bottom: ground-truth, S2GRU, RIMs, RMC, LSTM. Note that all models were trained on sequences with 3 bouncing balls, and the global state was reconstructed by stitching together $11 \times 11$ patches from the models (queried on a $4 \times 4$ grid).

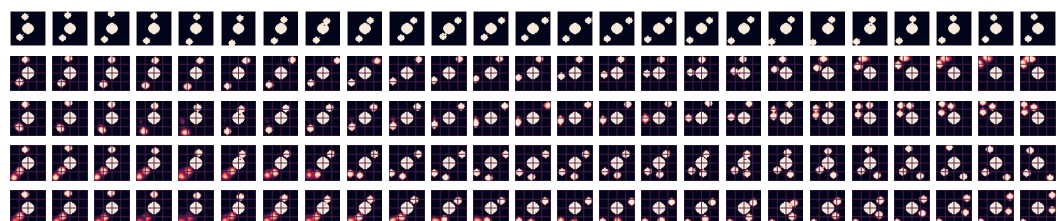

Figure 16: Rollouts (OOD) with 2 bouncing balls, from top to bottom: ground-truth, S2GRU, RIMs, RMC, LSTM. Note that all models were trained on sequences with 3 bouncing balls, and the global state was reconstructed by stitching together $11 \times 11$ patches from the models (queried on a $4 \times 4$ grid).

### E.4.4 OBJECTIVE FUNCTIONS

In the Starcraft2 task, predicting the next state entails predicting images of binary friendly markers, categorical unit type markers and real valued HECS markers. Accordingly, the loss function is a sum of a binary cross-entropy term (on friendly markers), a categorical cross-entropy term (on unit-type markers) and a mean squared error term (on HECS markers).

In the Bouncing Balls task, the model output is a binary image. Accordingly, we use a pixel-wise binary cross-entropy loss.

## F ADDITIONAL RESULTS

### F.1 BOUNCING BALLS

### F.1.1 ROLLOUTS

To obtain the rollouts in Figure 3, we adopt the following strategy. For the first 20 *prompt-steps*, we present all models with exactly the same $11 \times 11$ crops around randomly sampled pixel positions for 20 time-steps. For the next 25 steps, all models are queried at random pixel positions[10], and the resulting predictions (on crops) are thresholded at $0.5$ and fed back in to the model for the next step (at known pixel positions from the previous step).

Also at every time-step, the models are queried for their predictions on 16 pixel locations placed on a $4 \times 4$ grid. The resulting predictions are stitched together and shown in Figures 15, 16, 17, 18, 3 and 19.

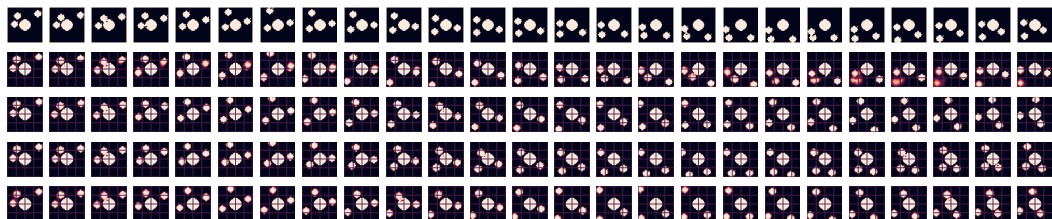

Figure 17: Rollouts (ID) with 3 bouncing balls, from top to bottom: ground-truth, S2GRU, RIMs, RMC, LSTM. Note that all models were trained on sequences with 3 bouncing balls, and the global state was reconstructed by stitching together $11 \times 11$ patches from the models (queried on a $4 \times 4$ grid).

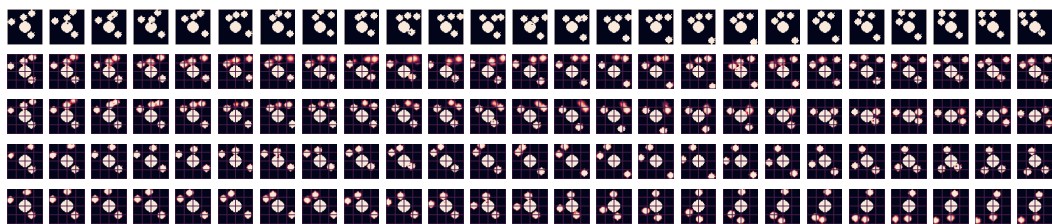

Figure 18: Rollouts (OOD) with 4 bouncing balls, from top to bottom: ground-truth, S2GRU, RIMs, RMC, LSTM. Note that all models were trained on sequences with 3 bouncing balls, and the global state was reconstructed by stitching together $11 \times 11$ patches from the models (queried on a $4 \times 4$ grid).

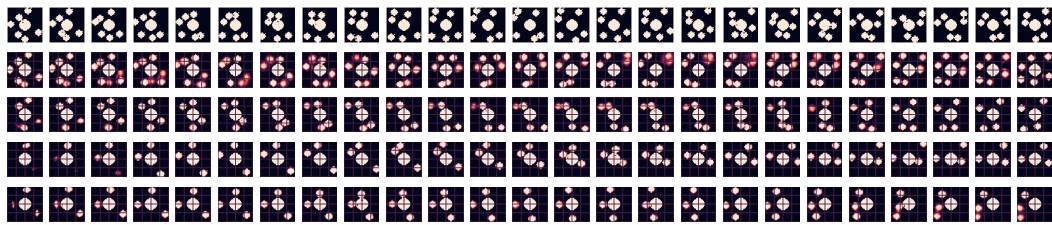

Figure 19: Rollouts (OOD) with 6 bouncing balls, from top to bottom: ground-truth, S2GRU, RIMs, RMC, LSTM. Note that all models were trained on sequences with 3 bouncing balls, and the global state was reconstructed by stitching together $11 \times 11$ patches from the models (queried on a $4 \times 4$ grid).

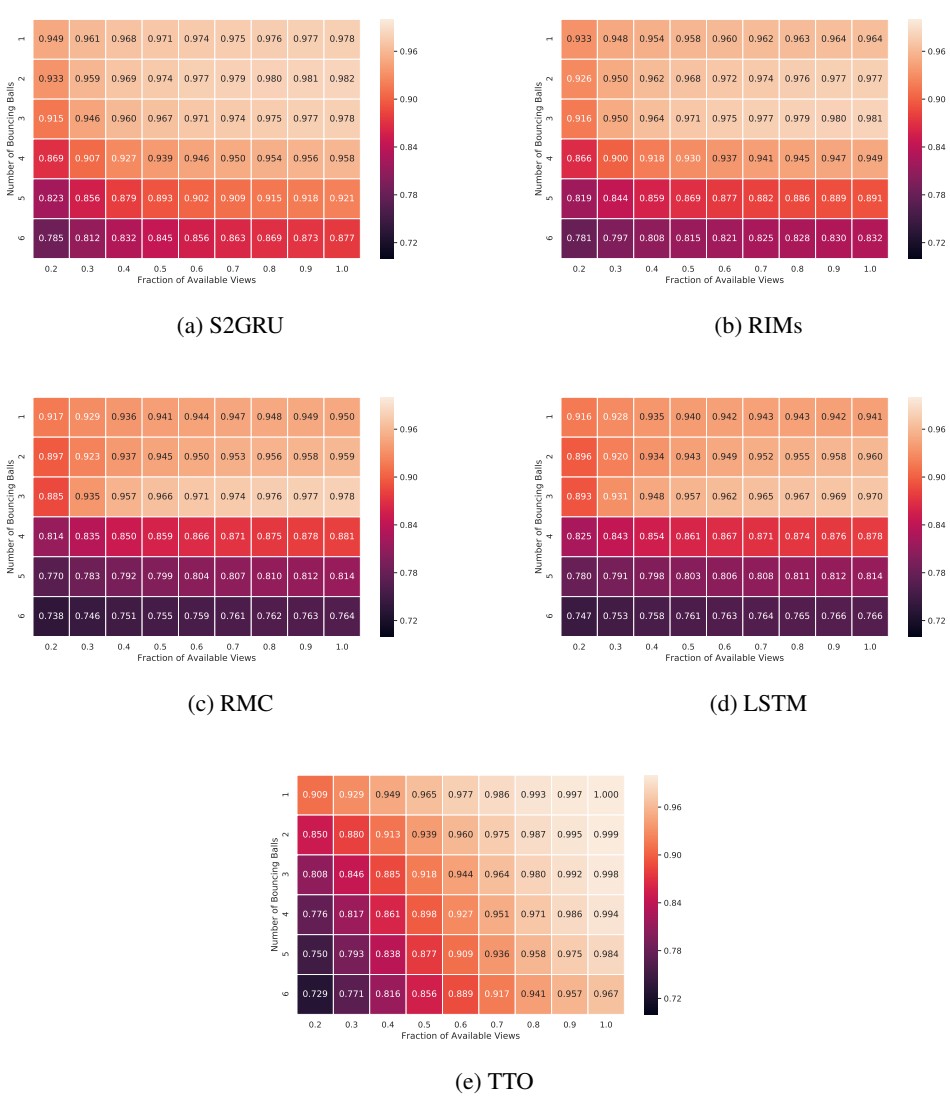

Figure 20: Balanced accuracy (arithmetic mean of recall and specificity) achieved by all evaluated models for one-step forward prediction task with various number of balls and fractions of available views. All models were trained on video sequences with 3 balls and a constant number of crops / views (10 views, corresponding to the right-most columns labelled 1.0). The color map is consistent across all plots.

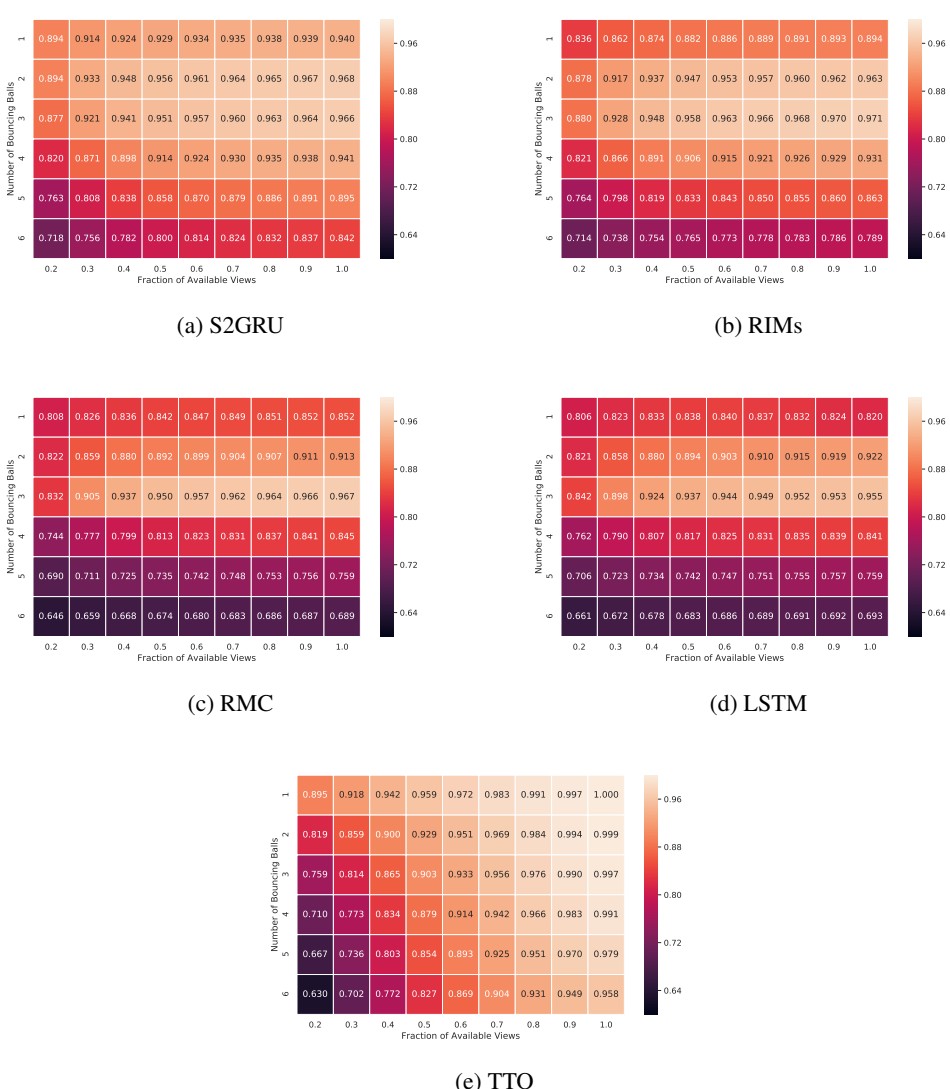

Figure 21: F1-Score (harmonic mean of precision and recall) achieved by all evaluated models for one-step forward prediction task with various number of balls and fractions of available views. All models were trained on video sequences with 3 balls and a constant number of crops / views (10 views, corresponding to the right-most columns labelled 1.0). The color map is consistent across all plots.

| Model
% of Active Agents | LSTM | RMC | S2GRU | S2RMC | TTO |
|---|---|---|---|---|---|
| 20% | 0.570565 | 0.586541 | **0.642292** | 0.637618 | 0.550806 |
| 30% | 0.599391 | 0.606114 | **0.660127** | 0.653950 | 0.578965 |
| 40% | 0.630606 | 0.640435 | **0.678752** | 0.671476 | 0.605867 |
| 50% | 0.638374 | 0.657472 | **0.688528** | 0.685988 | 0.627444 |
| 60% | 0.681040 | 0.704552 | **0.713851** | 0.708786 | 0.671961 |
| 70% | 0.709861 | **0.737436** | 0.734256 | 0.727980 | 0.723238 |
| 80% | 0.721041 | **0.748138** | 0.738611 | 0.732114 | 0.740936 |
| 90% | 0.750449 | **0.778647** | 0.755476 | 0.747613 | 0.786931 |
| 100% | 0.765592 | **0.795049** | 0.763126 | 0.754637 | 0.813504 |

Table 4: Friendly marker F1 scores on the validation set of the training distribution. Larger numbers are better.

| Model
% of Active Agents | LSTM | RMC | S2GRU | S2RMC | TTO |
|---|---|---|---|---|---|
| 20% | 0.323482 | 0.326685 | **0.435318** | 0.377538 | 0.297192 |
| 30% | 0.345108 | 0.350621 | **0.491934** | 0.433945 | 0.323736 |
| 40% | 0.373612 | 0.387733 | **0.540163** | 0.485278 | 0.350733 |
| 50% | 0.385550 | 0.406048 | **0.552589** | 0.510371 | 0.371088 |
| 60% | 0.430793 | 0.481986 | **0.599470** | 0.566149 | 0.435724 |
| 70% | 0.497964 | 0.590214 | **0.635928** | 0.606039 | 0.539652 |
| 80% | 0.579952 | 0.649277 | **0.650682** | 0.623040 | 0.617973 |
| 90% | 0.657643 | **0.694158** | 0.675294 | 0.655581 | 0.699008 |
| 100% | 0.677952 | **0.715929** | 0.689669 | 0.672186 | 0.737745 |

Table 5: Unit-type marker (macro averaged) F1 scores on the validation set of the training distribution. Larger numbers are better.

### F.1.2 ROBUSTNESS TO DROPPED VIEWS

In this section, we evaluate the robustness of all models to dropped crops on in-distribution and OOD data. We measure the performance metrics on one-step forward prediction task on all datasets (with 1-6 balls), albeit by dropping a given fraction of the available input observations.

Figure 20 and 21 visualize the performance of all evaluated models. We find that S2GRU maintains performance on OOD data even with fewer views (or crops) than it was trained on. Interestingly, we find that the time-travelling oracle (TTO), while robust OOD, is adversely affected by the number of available views. This could be because unlike the other models, it cannot leverage the temporal information to compensate for the missing observations.

### F.2 STARCRAFT2

### F.2.1 TABULAR RESULTS

The results used to plot Figure 8 can be found tabulated in Tables 4, 5, 6 and 7.

---

[10]These random pixel positions are the same for all models, but change between time-steps

| Model
% of Active Agents | LSTM | RMC | S2GRU | S2RMC | TTO |
|---|---|---|---|---|---|
| 20% | -0.014035 | -0.013569 | **-0.011491** | -0.011921 | -0.014174 |
| 30% | -0.013355 | -0.012747 | **-0.010631** | -0.011101 | -0.013539 |
| 40% | -0.012567 | -0.011808 | **-0.009906** | -0.010367 | -0.012916 |
| 50% | -0.012220 | -0.011305 | **-0.009637** | -0.009887 | -0.012481 |
| 60% | -0.010888 | -0.009799 | **-0.008751** | -0.009034 | -0.010929 |
| 70% | -0.009738 | -0.008469 | **-0.008068** | -0.008359 | -0.009184 |
| 80% | -0.009081 | -0.008027 | **-0.007873** | -0.008162 | -0.008466 |
| 90% | -0.007970 | **-0.007180** | -0.007347 | -0.007615 | -0.007038 |
| 100% | -0.007638 | **-0.006823** | -0.007103 | -0.007362 | -0.006401 |

Table 6: HECS Negative MSE on the validation set of the training distribution. Larger numbers are better.

| Model
% of Active Agents | LSTM | RMC | S2GRU | S2RMC | TTO |
|---|---|---|---|---|---|
| 20% | -0.303051 | -0.300892 | **-0.141989** | -0.146553 | -0.434099 |
| 30% | -0.258878 | -0.256878 | **-0.126037** | -0.137025 | -0.347899 |
| 40% | -0.216924 | -0.211048 | **-0.113317** | -0.126882 | -0.276596 |
| 50% | -0.206582 | -0.191644 | **-0.108643** | -0.113293 | -0.245019 |
| 60% | -0.158170 | -0.142643 | **-0.094380** | -0.099989 | -0.175233 |
| 70% | -0.126446 | -0.109634 | **-0.084129** | -0.089527 | -0.120694 |
| 80% | -0.111735 | -0.099229 | **-0.081624** | -0.086723 | -0.104135 |
| 90% | -0.082463 | -0.074518 | **-0.073439** | -0.078197 | -0.071243 |
| 100% | -0.070488 | **-0.063183** | -0.069856 | -0.074041 | -0.057276 |

Table 7: Log Likelihood (negative loss) on the validation set of the training distribution. Larger numbers are better.

