# OpenReview forum: "Spatially Structured Recurrent Modules"
_ICLR.cc/2021/Conference — ICLR 2021 Poster_

### Official Review · AnonReviewer4 · 2020-10-21
**Interesting problem setting and reasonable approach but important experiments are missing and I have concerns regarding the experimental results**

**Rating:** 6
**Confidence:** 4

**Review:**

### Summary

This paper is concerned with making predictions about the (global) state of a dynamical system in a partially observable setting, where only local observations are available. Concretely, this paper studies video prediction from glimpses and world-modeling in a multi-agent setting.

To solve this task the paper proposes Spatially Structured Recurrent Modules (S2RMs), an RNN-based dynamics model based on interacting sub-systems. S2RMs are an extension of RIMs (Goyal et al., 2019) with the main difference being that each module additionally contains an embedding vector that is meant to reflect its location in some learned metric space. This positional information can then be used to limit the interactions between modules, to distribute inputs (that come with an associated location) to the modules, and to query a particular subpart of the global state at a future point in time.

It is shown how S2RMs outperforms a number of baselines, which are obtained by combining the query mechanism of a GQN with either an LSTM, RIMs, or a Relational RNN to model the dynamics.

### Pro’s / Con’s / Justification

Overall I find that the paper is reasonably well written, although the clarity can be improved significantly. The first _four_ pages essentially only focus on motivation and problem statement, which is excessive, especially since the main contribution is an improved RNN architecture. In contrast, Section 4 that describes the method leaves important details about the adapted input attention mechanisms and inter-cell attention mechanisms to the appendix. Similarly, the experiments on the grid-world are entirely discussed in the Appendix, which I have therefore not considered as part of the contribution. More generally, the problem formulation seems unnecessarily broad given the relatively narrow contribution that is made.

The considered problem setting is interesting and I believe relevant to several real-world settings. The contribution itself is rather incremental since it essentially only involves associating a location with each module in RIMs. The adaptations to the existing attention mechanisms in RIMs that this then necessitates are straightforward, although additionally having the RIM input attention over the local observations is interesting and appears novel (I couldn't figure out if the original RIMs already attend to a set-representation of the input -- i.e. where each element is a different spatial location of the learned CNN representation). However, this paper does not compare alternative ways of implementing these changes or provide an ablation, which leaves it open what the effect is of adapting the attention mechanisms in this way. I also note that requiring the local observations to include an associated spatial location is arguably a more narrow setting than that explored in RIMs (if the input attention in RIMs is additionally directed to attend over the encoded local observations) and it would have been interesting to understand the importance of this. For example, how do S2RMs compare to RIMs when a similar encoder is used for RIMs (except for using the spatial coordinate)?

The experimental evaluation indicates that S2RMs outperforms the considered baselines and most notably RIMs, although there are some concerns. Firstly, for the baselines, the aggregated representations are computed as a sum of the output of the encoder applied to each local observation, which seems like a major bottleneck. While S2RMs can use attention to attend to each patch separately, information is almost certainly lost for the baselines in this way. I can understand why this may be necessary for the LSTM, but why not let RIM attend to the encoded local observations separately? Currently, this makes it difficult to understand in what way S2RMs improve over RIMs, especially since the reported margin already is quite small. Secondly, and related to this, I believe that it is important to include an ablation of the changes proposed to the attention mechanism. For example, what happens if positional information is only used for the recurrence, but not for the input attention (i.e. only using the global term)? Or what if it is only used for input attention, but not for the recurrence? Understanding the effect of these design choices would help strengthen the contribution and make it more significant compared to RIMs.

Given these issues I can not quite yet recommend an accept at this point in time, but I would be willing to increase my score depending on the outcome of some of the experiments I have suggested. More generally, I encourage the authors to revise Section 2-4 to put more emphasis on the actual contribution and discuss the specific design in detail.

### Detailed comments

* Why is it necessary to choose between the product of local and global terms and only the local term for the input attention? It seems to me that the local terms should suffice, and I wonder what the effect is of having this additional term.

* I would appreciate a more detailed discussion of prior work in the related work section. Currently, the paper only makes ‘sweeping claims’ about how the considered setting is different from a bunch of prior approaches. Rather, it would be more interesting to point out specific parallels, or discuss how ideas from prior work may be included in the considered set-up, or in what way those ideas have already been re-used.

* I am also missing a discussion of object-centric approaches to performing physical prediction tasks, like RNEM (Steenkiste et al., 2018), OP3 (Veerapaneni et al., 2020), etc. especially in relation to the considered bouncing balls task. Arguably, on this task these methods provide the best trade-off between locally interacting sub-systems and more global modeling since each RNN specializes to a specific object, which is evidently not achieved by S2RMs. Since each RNN learns to specialize on a single object, it can easily be viewed as modeling the global state of the system through locally interacting subsystems acting on partial observation. On the other hand, it is clear that S2RMs (and RIMs by extension) offer other advantages. For example, an advantage of S2RMs is that they consider modules having their own weights, which can thereby specialize on specific interactions between objects (or break down interactions between objects further). RNEM and the likes are limited to modeling the same set of interactions between interacting subsystems given by objects, although evidently, the notion of an object can also be flexible in this case (i.e. when an object-centric representation corresponds to two physical objects in pixel space).

* The visualization in Figure 5, suggests that individual modules specialize, but there also seems to be quite some redundancy. Would it not be possible to quantify the achieved modularity somehow, eg. by measuring IoU? In that case it would be interesting to see how the system behaves when the number of provided modules is insufficient to model each individual actor in the considered system, and when the number of modules is equal to the number of actors (eg objects on bouncing balls). Additionally, do you have any intuition for what happens if modules are removed at test-time?

* Why are RIMs not included in Table 1? I would be surprised if it performs that much worse to S2RM, since both models perform similar on the bouncing balls task. If this is the case then please at the very least provide some intuition or insight to explain this difference.

* Please include a discussion of limitations in the conclusion.

* Please take a moment to go through the references and correctly cite papers that have been published.

%%%%%%%%%%%%%%%%%%%%%%%%%%%%%%%%%%%%%%%%%%

I have improved my score following the improvements made by the authors. See my reply below for details.

---

> ### Author Response · Authors · 2020-11-15
> **First response to R4 (part 1): on novelty w.r.t. RIMs and adequacy of the baseline encoder**
>
> Hi R4, thank you for your detailed review! We are glad that you find our problem setting interesting and relevant to several real-world settings, and appreciate your comments and suggestions, especially regarding the writing -- we had prioritized an intuitive exposition over mathematical details (as recommended by a previous reviewer), but we will find a better balance and update the manuscript in the coming days. We will also attempt to run as many of your suggested experiments as possible (subject to constraints of time and resources) and report back with the results.
>
> For now, we begin by addressing your concerns about the delta to RIMs [1]. While it is indeed the case that RIMs and S2RMs (along with several other works) share the general paradigm of independent modules that interact via the bottleneck of attention, we enumerate several dimensions along which S2RMs and RIMs significantly differ.
>
> - **The way sparsity is induced:** while both RIMs and S2RMs have modules interact via sparse attention, the motivations are fundamentally orthogonal. In S2RMs, we exploit the spatial structure of the environment to induce sparsity -- if modules A and B are responsible for different and independent spatial regions of the environment, they might learn to not interact with each other. The sparse attention mechanism in RIMs on the other hand is inspired by the notion of competition between experts [2] -- at every time step, the $k$ RIMs that have the largest attention weights to the (dynamically changing) inputs are activated. These active RIMs can now read from all other RIMs (active or inactive), but the states of inactive RIMs are not updated. S2RMs have no such notion of competition: if the learned embeddings of module A is similar to that of module B, they may both attend to similar spatial regions and therefore choose to collaborate.
> - **Module activation and the notion of default dynamics:** In RIMs, modules that are not activated at a time-step are essentially replaced by identities, i.e. they propagate their state forward in time as is without evolving them. In S2RMs, there is no notion of module activation. If no observations can be addressed to a module, then the module is fed a zero vector but it may still evolve its hidden state (e.g. in order to keep track of entities that are evolving but are not observed). In particular, the two modules may still continue interacting with each other even though neither is fed an input, because we assume that the underlying system might evolve even if no observations are available.
> - **Stability of _named_ modules:** Each module in S2RMs are associated with an embedding vector (or a _name_) that determines the region of space in the environment it is responsible for modelling. This yields stability in what the modules expecte to receive as inputs and enables them to specialize to a region in space. This is unlike in RIMs, where there is no notion of _named modules_, and the input to a module (i.e. whether it is available or not) is contingent on the activations of all other modules.
> - **State and Location-based vs. purely state-based interactions:** In S2RMs, whether or not two modules interact is a function of both their embeddings (which also controls what locations they are responsible for modelling) and their states. In RIMs, the interaction between modules is contingent only on their states. We therefore use spatial information in a way RIMs doesn't.
>
> >Firstly, for the baselines, the aggregated representations are computed as a sum of the output of the encoder applied to each local observation, which seems like a major bottleneck...
>
> We did in fact take special care to verify that this is not the case, as can be seen by the stellar in- and out-of-distribution performance of the time-travelling oracle (TTO).
>
> At time $t$, TTO is provided with (priviledged) ground-truth local views at time $t + 1$ -- these are then aggregated with the additive scheme (as in all other baselines) and the resulting representation is queried at different locations, also at time $t + 1$. In other words, TTO essentially operates in an auto-encoding setting, but with additive aggregation applied to the intermediate representations (similar to the original GQN architecture). Figure 4 clearly shows that this scheme can acheive excellent generalization -- far better than S2RMs or any other baselines -- confirming that the encoding and aggregation scheme can handle the partial observability and is not the bottleneck for generalization. This also justifies the use of this architectural scaffolding to test various recurrent models that are not designed with the partially observable setting in mind.
>
> **Continued in part 2.**
>
> ---------- REFERENCES ----------
>
> - [1] https://arxiv.org/abs/1909.10893
> - [2] http://papers.neurips.cc/paper/157-on-the-k-winners-take-all-network.pdf

---

> > ### Author Response · Authors · 2020-11-15
> > **First response to R4 (part 2): on the local and global terms in the attention mechanism and modularity**
> >
> > **Continued from part 1 (parent comment).**
> >
> > > I can understand why this may be necessary for the LSTM, but why not let RIM attend to the encoded local observations separately? Currently, this makes it difficult to understand in what way S2RMs improve over RIMs, especially since the reported margin already is quite small.
> >
> > In light of the discussion above (concerning the adequacy of the architectural scaffolding), we see it justified to use RIMs as a drop-in replacement for LSTMs, as suggested by the authors of RIMs (cf. Section 2.4 and Section 4.3 of [1]). Further, it is difficult for us to predict how the additional attention mechanism would interact with the many other components in RIMs, and extensive modifications to a baseline (which entails a thorough code-review from the original authors and a comprehensive hyperparameter sweep) is beyond the scope of this paper.
> >
> > > Secondly, and related to this, I believe that it is important to include an ablation of the changes proposed to the attention mechanism.
> >
> > Thank you for the suggestion! We are running a suite of ablations and will report back with the rsults.
> >
> > > Why is it necessary to choose between the product of local and global terms and only the local term for the input attention? It seems to me that the local terms should suffice, and I wonder what the effect is of having this additional term.
> >
> > The purpose of the global term is to modulate the attention based on content, and not just location. For instance, even if the local term permits interaction between modules A and B, the global term can override this and choose to have A and B not interact conditioned on the hidden states of the respective modules. This can be useful e.g. if two modules focus on the same region in space, but specialize to different aspects of the dynamics.
> >
> > On the contrary, if the local term does not permit interaction between A and B, then they do not interact irrespective of the global term. Likewise for the inputs: even if the local term allows for an input to be addressed to a module, the non-local term allows the module to reject it based on its content. In this spirit, the product of the local and global term allows for more sparsity than the local term alone.
> >
> > > I am also missing a discussion of object-centric approaches to performing physical prediction tasks...
> >
> > Thank you for pointing this out. The setting explored in object-centric approaches is fundamentally different from ours in that we do not consider an object as a sub-system. Instead, we learn to spatially partition the environment such that each module (RNN) is responsible for all objects or agents its _territory_ (or _enclave_). Further, given that we do not rely on entity-grounding, our approach might be applicable in settings where object-centric methods are not, e.g. modelling fluid motion from partial observations.
> >
> > > The visualization in Figure 5, suggests that individual modules specialize, but there also seems to be quite some redundancy. Would it not be possible to quantify the achieved modularity somehow, eg. by measuring IoU? ...
> >
> > We do not suspect that the overlap in spatial enclaves (i.e. some redundancy) is detrimental. For instance, if a region of the environment is challenging, one might expect multiple modules to "team up" and share capacity (i.e. they might specialize to different things happening in the same location). Likewise, we might expect certain modules to capture a broad overview of the environment whereas other modules focus on smaller pockets of activity in the environment.
> >
> > Moreover, an IoU based measure of modularity would only reflect "location-based" modularity (due to the local term), but not "content-based" modularity (due to the global term).
> >
> > **Continued in part 3.**
> >
> > ---------- REFERENCES ----------
> >
> > - [1] https://arxiv.org/abs/1909.10893
> > - [2] http://papers.neurips.cc/paper/157-on-the-k-winners-take-all-network.pdf

---

> > > ### Author Response · Authors · 2020-11-15
> > > **First response to R4 (part 3)**
> > >
> > > **Continued from part 2 (parent comment)**
> > >
> > > >  Additionally, do you have any intuition for what happens if modules are removed at test-time?
> > >
> > > This is an interesting question! We will investigate what happens and report back.
> > >
> > > > Why are RIMs not included in Table 1? I would be surprised if it performs that much worse to S2RM, since both models perform similar on the bouncing balls task. If this is the case then please at the very least provide some intuition or insight to explain this difference.
> > >
> > > We were unable to perform well on the validation set with RIMs, even after extensive parameter tuning. Given that we only include baselines that perform at least as well as S2RMs (or better) on the validation set, we did not consider it for evaluation.
> > >
> > > As to why this is the case is an interesting question, and our preliminary hypothesis is the following. We read from Figure 6 that RMC (Relational Memory Cores) is the best performing model on the validation set (i.e. at agent drop probability = 0), indicating that the task aligns well with the inductive bias of a relational memory. This is intuitive because unlike in the Bouncing Balls environment where a few frames might be enough to predict the dynamics for the next few, SC2 is a "fast-moving" environment potentially requiring instant communication between observers. This suggests that a fast communication channel between observers (e.g. in form of a memory that all observers have read/write access to as in RMC) can help. Unlike in RIMs, S2RMs maintain communication between modules even when they are not provided with inputs, which could explain why S2RMs perform better than RIMs on the validation set.
> > >
> > > This concludes our first response while the ablation experiments run. In the mean time, if we did not answer something to your satisfaction, please feel invited to leave a comment and engage with us!
> > >
> > > ---------- REFERENCES ----------
> > >
> > > - [1] https://arxiv.org/abs/1909.10893
> > > - [2] http://papers.neurips.cc/paper/157-on-the-k-winners-take-all-network.pdf

---

> ### Author Response · Authors · 2020-11-20
> **Second Response to R4: manuscript updated to reflect your suggestions**
>
> Hello R4, we once again thank you for your comprehensive review. We have updated our manuscript to reflect your suggestions about adding ablation experiments and revising sections 2-4. We have also expanded on the related work to clarify the differences between the proposed method and prior work, and include a discussion about the limitations and avenues of future work (please see the [top-level comment for a detailed change log](https://openreview.net/forum?id=5l9zj5G7vDY&noteId=PJV-j1Filh)). If there are further concerns of yours that we are yet address in our first response or the update, please do not hesitate to let us know!
>
> Below, we complement our previous response with new insights we have gained from the additional experiments.
>
> > Why is it necessary to choose between the product of local and global terms and only the local term for the input attention? It seems to me that the local terms should suffice, and I wonder what the effect is of having this additional term.
>
> Our ablation experiments show that both local and the non-local terms contribute to the final performance. The non-local term is particularly important for input attention, whereas the local term is crucial for the inter-cell attention. These results are compatible with intuition that modules filter their inputs based not only on location but also on content.
>
> > Additionally, do you have any intuition for what happens if modules are removed at test-time?
>
> Thank you again for the suggestion, we now include this experiment. In summary, the performance degrades gracefully as modules are removed at test-time, suggesting that the co-adaptation between modules is limited.

---

### Official Review · AnonReviewer2 · 2020-10-28
**A really interesting architecture. Evaluation tasks could be more challenging. Ablation is missing. Clarity of the paper could be improved.**

**Rating:** 7
**Confidence:** 4

**Review:**


### Summary of the paper

The authors propose a novel architecture for synthesizing information from multiple local observations and making predictions into the future.
Given a set of observations, each with an associated location, the model maps these into an embedding space.
A set of RNNs, each with a learned embedding that corresponds to a vector in the embedding space, is used to process observations that are close in the embedding space.
The activations that the RNNs can receive, exchange between each other and output for a given query location are modulated by the distance in the embedding space.
This is achieved through three separate attention mechanisms with a similar design: A fully non-local (all-to-all) attention layer is followed by a multiplication with a kernel that falls off (and is eventually cut off to 0) with distance.
This type of attention is used once for computing the inputs to the RNNs, once for the hidden state of the LSTM (exchanging information between RNN modules that are close by) and once when computing an output response given a query location.

### Relation to prior work

The paper references previous work that deals with synthesizing information from multiple localized observation and positions itself reasonably with respect to prior work.

### Experimental evaluation

The authors demonstrate results on two domains: A bouncing balls video prediction task which is well-known and useful but could be considered a toy task and predicting sequences from StarCraft 2 battles. In the case of StarCraft 2 the target for the prediction are raw unit stats and actions rather than images.
The model generally outperforms baselines on the bouncing balls task and generalizes better on the StarCraft 2 domain (but does not outperform baselines on the training task itself).
It's difficult for me to judge whether the model will also work on other domains based on these experiments.

I'm curious whether some of the components of the model are actually needed (e.g. attention between RNN modules). In my opinion it would improve the paper significantly if some results on ablation experiments would be provided. Another aspect of the model that could be investigated is the dependence of the performance on the kernel that modulates information exchange by distance in the embedding space. What happens if the modulation is disabled?

### Presentation and clarity

The paper is generally well written. But I did have some difficulties understanding the goal and approach because the paper relies on a lot of wordy exposition (some of which is highly speculative in my opinion). I believe that the paper would be clearer if the introduction was shortened. This could also leave more space to discuss the experiments in more detail or add an ablation study.

### Conclusions

I found the architecture introduced in the paper genuinely interesting and would like to see more work in this direction. I would give the paper a higher rating if it included ablation experiments, or if it was adjusted to simplify and shorten some of the discussion in the introduction.


==========

Edit after author comments:
I've read the author comments and the updated version of the paper.
Although the authors claim that they have shortened the introduction of the paper by 1 page, this doesn't actually seem to be the case in the last version that was uploaded, where the section titled "Introduction" is almost unchanged compared to the original upload (I've used the diff tool between the latest and original version).
Maybe there was a misunderstanding and the authors have shortened a different part of the paper?
Although it would have been nice to shorten and streamline the introduction to make the paper easier to read, it's not critical to my rating.
The added ablation experiments demonstrate that each of the different attention modules proposed in the paper improve results, which I think really improves the paper. I was originally not sure whether the complexity of the model was justified, but the new experiments demonstrate that each of the components seems to be needed.
I've also carefully read the rest of the paper and the author comments explaining details of the tasks under study and now feel that I have a much better understanding of what was done and how the model could be used for other tasks.
I agree with R4 that the novelty of the paper might not be groundbreaking, but I believe the paper could be relevant and interesting for other researchers who want to incorporate attention mechanisms into their architectures, so I recommend accepting the paper.
I've increased my rating from 6 to 7.

---

> ### Author Response · Authors · 2020-11-15
> **First response to R2: on challenging aspects of the evaluation tasks**
>
> Hello R2, thank you for your review! We're glad that you find our architecture exciting and our paper well written. We appreciate your feedback about the lengthy introduction, and we will update the manuscript in the coming days to reflect your suggestion. We are also running a suite of ablation experiments; we will report back with the results. In the mean time, we respond to your concerns below.
>
> > A bouncing balls video prediction task which is well-known and useful but could be considered a toy task...
>
> We would like to point out that unlike the classical bouncing-balls video-prediction task, we only work with local crops of the video frames. In other words, given a set of small crops of the video frame (along with the location of the central pixel) at time $t$, the task is to output a crop around an arbitrary query pixel (that is a priori unknown) at time $t + 1$. The problem is set up such that at any time, the model has access to at most 52% of the frame at a time (which is a very unlikely case) -- on average, this number is around 33% of the frame.
>
> This makes the problem significantly more challenging than the typically studied setting where the entire video frame is available to the model. This is because the model must now place the available crops in appropriate spatial context order to build and maintain a consistent representation of the latent global state (i.e. the entire video frames).
>
> > In the case of StarCraft 2 the target for the prediction are raw unit stats and actions rather than images.
>
> Our model operates on "semantic images", akin to segmentation masks in computer vision. Indeed, the goal is not to solve the computer-vision task of recognizing the units from rendered images, but to model the dynamics of the environment with rich and complex rules from local observations. Moreover, the use of semantic information is fairly common in this domain (see e.g. `detailed-architecture.txt` in the supplementary material of AlphaStar [1] or the section on "State and Observations" in [2]) and we adopt this problem setting.
>
> > It's difficult for me to judge whether the model will also work on other domains based on these experiments.
>
> Intuitively, we expect the proposed method to shine in spatial domains that (a) only afford a relatively small number of localized views in to itself and (b) are sparsely active, i.e. larger parts of the environment are relatively inactive but there are certain spatial "hotspots" of activity (that are not known a priori). The modules could then learn to arrange themselves in a way that such hotspots are assigned a cluster of modules (to augment processing capacity), whereas the inactive regions are handled by a smallar number of modules in order to not waste capacity.
>
> > In my opinion it would improve the paper significantly if some results on ablation experiments would be provided. Another aspect of the model that could be investigated is the dependence of the performance on the kernel that modulates information exchange by distance in the embedding space. What happens if the modulation is disabled?
>
> Thank you for the suggestion! We will report back with a suite of ablation experiments.
>
> This concludes our first response. In case something is left unclear, please do not hesitate to interact with us!
>
> ---------- REFERENCES ----------
>
> - [1] https://www.nature.com/articles/s41586-019-1724-z
> - [2] https://arxiv.org/abs/1902.04043

---

> ### Author Response · Authors · 2020-11-20
> **Second response to R2: we have updated the manuscript to streamline the introductory sections and add ablation experiments**
>
> Hi R2, we thank you again for your review! We have updated our manuscript to reflect your suggestions: (a) the introductory sections are now much more concise (a page shorter) and (b) we now include a suite of ablation experiments over the components of the attention mechanism (please see the [top-level comment for a full changelog](https://openreview.net/forum?id=5l9zj5G7vDY&noteId=PJV-j1Filh)). If there is something that we are yet to address to your satisfaction, please feel invited to get in touch with us!

---

### Official Review · AnonReviewer3 · 2020-10-29
**Learning shared topological structure for partially observed data: an interesting idea with wider applications**

**Rating:** 7
**Confidence:** 4

**Review:**

The authors propose a recurrent state space model for partially observed data, where at any given time, only a partial subset of observations are accessible to the model to make future predictions. This is an important and challenging problem in ML as we often only have access to a partial view of temporally evolving data.

The authors propose uses a recurrent model, which models a dynamically evolving partially observed process by introducing spatio-temporal interactions across multiple recurrent neural nets. The key contribution of their work is the use of a shared embedding space, that allows them to employ a mechanism to modulate spatial and temporal interactions between the RNNs.

The proposed method learns a mapping function for both, the embeddings corresponding to the RNNs, and the embedding corresponding to the observations and locations, to a shared metric space. Since the functions used to map the observations and locations, and the embeddings vectors associated with each RNN, are learned, the joint learning objecting corresponds to learning a topological structure on the points in this shared metric space. Additionally, the authors propose a truncated kernel to attenuate effects of observations far apart in time.

The final learning objective then encourages the shared metric space to be topologically organized in a way such that the embedding space is partitioned into regions, where each region places a varying “responsibility” over each of the different RNNs. Since the observations are also embedded into this space, any given observation can then be mapped into the embedding space, in the embedding space, the observation embedding be close to one or more embedding points corresponding different RNNs (which in turn models the dynamics in that regime).

The idea itself is quite interesting and allows for a “soft” representation of a state space model which through the embedding respace resembles an explicit duration + factorial Markov model. The shared embedding space is an interesting idea and can be extended to other time series models as well.

The experiments along comparisons to strong baselines demonstrates the validity of their approach. In addition to the interesting core idea, the authors employ a fairly involved attention mechanism. Given the space limitation it is understandable, however, it would have been informative to have some experiments with some ablation studies i.e. what is the simplest model structure they could have used while still incorporating the shared metric space idea to make the model more parameter efficient.


Some minor notational issues:
D_x seems to be defined as a function space in the model, however, it is not a function space. One could think of the forward evolutions of an RNN with a memory cell having an equal representation in function space, however, I feel the function space characterization of dynamical system section 2 obfuscates the model structure. Additionally, s_t^a can be inferred to be P(x_t^a), however, s_t^a is not defined in the text.


A few clarifications from the textual descriptions:
Could the authors elaborate on how exactly does E(.) process all observations in parallel across t and a?

“Explained by the fact that unlike recurrent models, it does not leverage the temporal dynamics to fill in the missing information due to fewer available observations.” Could you elaborate on why you don’t expect the individual RNNs to leverage the temporal dynamics?

In the experiments, does the reported metric F-1 score account for label switching?

As currently modeled, only the forward rollout of the RNNs are used, could this be extended to having a Bi-directional RNN – one might expect better global estimates as we switch from “filtering” to a “smoothing” estimate

Overall, I believe this an interesting research direction and the approach proposed by the authors can be extended to other useful models. The experiments are well thought out. The paper organization could be clearer but as it stands it is easy to understand after a thorough read.

---

> ### Author Response · Authors · 2020-11-14
> **First response to R3 (part 1)**
>
> Hello R3, thank you for an encouraging review -- we are excited that you find our idea interesting, and our experiments well thought out! The paper organization could indeed be clearer, and this is one of the things we are improving in an updated revision that we will upload in the coming days.
>
> > it would have been informative to have some experiments with some ablation studies i.e. what is the simplest model structure they could have used while still incorporating the shared metric space idea to make the model more parameter efficient.
>
> This is indeed an interesting question, thank you for the suggestion! A suite of ablation experiments are underway and we will include them in the updated manuscript.
>
> > Some minor notational issues: D_x seems to be defined as a function space in the model, however, it is not a function space. One could think of the forward evolutions of an RNN with a memory cell having an equal representation in function space, however, I feel the function space characterization of dynamical system section 2 obfuscates the model structure.
>
> **Note:** OpenReview's Latex parsing seems to be buggy as of writing this post, which is why we have replaced \mathfrak{O} with D and \mathfrak{o} with o.
>
> Thank you for this feedback, we will improve our explanation in the next revision. In intuitive terms, we think of $D_{\mathcal{X}}$ as the set of all "world-states", and the actual partial observations $\mathbf{O}$ result from querying a world-state $o \in D_{\mathcal{X}}$ at a known location $\mathbf{x}$, i.e. $\mathbf{O} = o(\mathbf{x})$. Here, by world-state we mean the state of the entire environment (i.e. the state had it been fully observable), which we do not assume to observe, but represent as a mapping from $\mathbf{x}$ to $\mathbf{O}$. In particular, by $D_{\mathcal{X}}$ we do not mean the space of functions that the RNNs themselves may represent.
>
> Nevertheless, the goal of the proposed model is to infer and model the dynamics of the world-state $o_t$ in time $t$. In particular, we place no explicit constraint on where the model can be queried: given observations samples $(\mathbf{x}_i, o_t(\mathbf{x}_i))$, our model first implicitly constructs a representation $\hat o_t$, evolves it in time to represent $\hat{o}_{t + 1}$, and then answers queries $(\mathbf{x}_j)_j$ to obtain $(\mathbf{x}_j, \hat{o}(\mathbf{x}_j))$. Finally, the objective function matches $\hat{o}(\mathbf{x}_j)$ with $o(\mathbf{x}_j)$ to ensure that $o$ and $\hat{o}$ align.
>
> >  Could the authors elaborate on how exactly does E(.) process all observations in parallel across t and a?
>
> We mean that the encoder $E$ does not mix information between $t$ and $a$, and processes each $x_t^a$ (for different $t$ and $a$) as though they were different samples.
>
> Consider that the observations are packed as a tensor of shape $(N, T, A, C, H, W)$, where $N$ is the number of sequences in the batch, $T$ is the number of time steps, $A$ is the number of observations, $C$ is the number of channels, $H$ and $W$ is the height and width (respectively) of the observations (when encoded as images). In order to obtain the encoded representations, the encoder $E$ first folds the $N$, $T$ and $A$ axes together to obtain a tensor of shape $(N \times T \times A, C, H, W)$. The network than processes each of the $N \times T \times A$ images separately to obtain a $D$-dimensional representation of shape $(N \times T \times A, D)$, which is then reshaped to have the shape $(N, T, A, D)$.
>
> > “Explained by the fact that unlike recurrent models, it does not leverage the temporal dynamics to fill in the missing information due to fewer available observations.”
>
> Thank you for asking, we will improve the phrasing on this sentence.
>
> We were referring to the fact that time-travelling oracle (TTO) does not learn to leverage temporal dynamics, because it is essentially "spoon-fed" the ground-truth future states (in other words, it has priviledged access to $\mathfrak{o}_{t + 1}$ at time $t$). Consequently, it is rendered helpless as the number of partial views available to it is reduced. All recurrent models, on the other hand, are capable of learning to leverage temporal dynamics (i.e. by gathering information over time). Accordingly, as the number of views available at a time is reduced, they are affected to lesser extent (S2GRUs and RIMs less than RMC and LSTM).
>
> **Continued in part 2.**

---

> > ### Author Response · Authors · 2020-11-14
> > **First response to R3 (part 2)**
> >
> > **Continued from part 1 (parent comment)**
> >
> > > In the experiments, does the reported metric F-1 score account for label switching?
> >
> > For the bouncing-balls experiments, the prediction problem at hand can be cast as a binary segmentation problem, i.e. a pixel-wise binary classification problem. In this case, we use the F1 score as the harmonic mean of (binary) precision and recall.
> >
> > For the Starcraft2 experiments, one of the prediction problems at hand is pixel-wise multiclass classification (analogous to semantic segmentation in the CV literature) for classifying the type of unit occupying a position in the agents' field of view. In this case, we use the macro-averaged F1 score, meaning that we compute the F1 score of each class individually and then average them with equal weights for all classes. This metric penalizes models that only perform well on frequent classes but poorly on infrequent classes.
> >
> > Since the class labels have semantic meaning and cannot be permuted (i.e. we do not have a clustering problem at hand), we do not account for label switching.
> >
> > > As currently modeled, only the forward rollout of the RNNs are used, could this be extended to having a Bi-directional RNN – one might expect better global estimates as we switch from “filtering” to a “smoothing” estimate
> >
> > This is indeed an exciting direction to pursue! In addition to bi-directional RNNs, one might also think about ways of encoding the combination of modular and spatial inductive biases for parallel-in-time architectures (e.g. neural processes and universal transformers) in order to obtain smoothing estimates.
> >
> > This concludes our first response. If there is something you think we could explain better or some aspect of your comment that we did not correctly understand, please do not hesitate to leave us comment!

---

> ### Author Response · Authors · 2020-11-20
> **Second response to R3**
>
> Hello R3, we thank you again for your positive review! Following your suggestion, we have updated our manuscript to improve the paper organization ([full change log in this top-level comment](https://openreview.net/forum?id=5l9zj5G7vDY&noteId=PJV-j1Filh)). We hope to have answered your questions in our first response; if this is not the case, please do not hesitate to engage with us!

---

### Official Review · AnonReviewer1 · 2020-10-29
**Interesting idea, hard to assess the importance of the proposed spatial localization.**

**Rating:** 6
**Confidence:** 3

**Review:**

This paper models noisy observations from complex dynamical systems, consisting of multiple interacting subsystems, by a set of sparsely interacting recurrent networks. The interactions between the recurrent networks (or modules) are constrained to be spatially localized, this is done by embedding the position of each module in a metric space and scaling the strength of the interactions between modules using this metric.

The overall effect of this constraint is to induce spatial structure in the global dynamics of the interacting modules. The paper argues that this constraint is a good inductive bias for capturing the dynamics of systems that consist of sparsely interacting agents, like cars and traffic lights in a traffic simulation, or characters and actions in a video game.

The paper uses recurrent neural networks for modeling the individual sub-systems, as these are flexible nonlinear dynamical systems that can model the (local) dynamics of individual modules. The interactions between these systems is captured through an attention mechanism. These attention weights are modulated (scaled) by a similarity between embedded positions of pairs of modules.

I found the paper interesting, but I thought the presentation could be clearer, and the paper could better demonstrate the importance of the spatial structure.

First, as far as I can tell, the main difference between this work and previous work (specifically, the recurrent interacting modules (RIM) paper) is the addition of the positional embedding and corresponding weighted interaction terms.

Given that the key novelty is the positional embedding, I think the paper should spend more time introducing and motivating how that is implemented, and experiments testing that implementation.

The positional embedding is introduced in a few lines at the top of section 4, with the motivation that the chosen function is "commonly used" (and cites the original transformer paper). However, the motivation for positional encodings in NLP tasks for the transformer seems to me to be completely different from the motivation for positional or spatial embeddings in this work. This work is (largely) aimed at modeling interacting systems in the physical world, whereas the positional encoding in Vaswani et al was motivated as a simple way to introduce signals to the network that represented position. Given that these are different domains, I think the positional embedding should be better motivate here. In addition, the paper suggests that "other choices might also be viable". What are these other choices? Did the authors experiment with them?

Second, how are the hyperparameters for the similarity metric (epsilon and tau) chosen? These govern how close two subsystems need to be to interact, but as far as I can tell they are simply presented as constants in a table in the appendix. How does performance vary as you change these? At the very least, the main text should state how these were chosen.

Third, from what I can tell, the S2GRU has less capacity than a system that models all of the interactions, such as a global LSTM. If so, then how come the LSTM underperforms on simple tasks such as the bouncing balls? Presumably, the LSTM could just learn the same interactions that are in the S2GRU. Is this underperformance due to issues with trainability (the global LSTM is harder to train?) or perhaps it still has limited capacity (e.g. it does not have enough units or layers to model the dynamics?). If the latter, it would be nice to see experiments with different numbers of units and/or more training data, showing that for large systems the LSTM is as good as (or better) than the S2GRU (since we expect the LSTM to be able to learn any nonlinear dynamics). Then, as you reduce either the number of units or training data, perhaps the S2GRU starts to outperform (as the inductive bias of the S2GRU starts to win). Basically, if the spatial structure is really an inductive bias, I would expect the benefit to go away with a higher capacity model or more training data (where inductive biases are not needed, as we can just learn directly from data).

Fourth, are there cases where the spatial structure is *not* a good inductive bias? For example, situations where modeling the interactions between all pairs of agents or subsystems is necessary to capture the dynamics. If so, I would like to see experiments on these problems that show that the S2GRU does *not* outperform a global model such as an LSTM. This would be a nice control to show that the limits of the inductive biases of the imposed spatial structure.

---

> ### Author Response · Authors · 2020-11-13
> **First response to R1 (part 1): intuition behind the positional embedding scheme**
>
> Hi R1, thank you for your review! We're glad that you found our paper interesting and we appreciate your feedback regarding presentation, especially the part concerning the motivation behind using positional embeddings. We will update our manuscript in the coming days to reflect your suggestions, but in the mean time you will find below our first response to your review.
>
> > The positional embedding is introduced in a few lines at the top of section 4, with the motivation that the chosen function is "commonly used" (and cites the original transformer paper)...
>
> The positional embedding is indeed an important component of our architecture, and we note that the functional form of the embedding that we use finds application beyond transformers in NLPs, for instance in 3D scene representation [2, 3] and protein structure modelling [6].
>
> In our case, the choice of a positional embedding determines a function-space of spatial functions that the local attention can represent (this is related to the functional form of the enclaves, as visualized in Figure 5). To see how, consider the local weight $w(x)$ with which module with embedding $\mathbf{p}$ may attend to an observation at location $x$, given by the inner product $\mathbf{p} \cdot P(x)$. We have that $w(x) = \sum_i (p_{2i} \cos(\omega_{i} x) + p_{2i + 1} \sin(\omega_{i} x))$ where $p_{j}$ are learnable parameters (for all $j = 0, ..., 2i - 1$) corresponding to frequencies $\omega_i$. Now, if we increase the embedding dimension to approach infinity (by increasing the number of frequencies $\omega_i$), we gradually recover the Fourier basis of $L^2$, the space of squared integrable functions. In this limit, we can recover all $L^2$ functions of $x$ that are normalized to a constant. In other words, our scheme can in principle learn to connect any two spatial locations $u$ and $v$ to the same module by learning a function $w$ such that $w(u) > \tau$ and $w(v) > \tau$ (where $\tau$ is an appropriate truncation parameter), yielding the model a lot of flexibility to learn any spatial structure / topology.
>
> But of course, infinitely many $\omega_i$ is not computationally feasible; we therefore sample $\log \omega_i$ on a grid, as do works before us [2, 6] and is justified in the theory of RKHS [1, 4]. In fact, we did experiment with the number of frequencies and found that including too many frequencies hurts the training. We have also experimented with learning $\omega_i$, but without success. These findings support the hypothesis that excessive flexiblity in terms of spatial structure might be detrimental to training performance. Another option that might worth exploring in future work could be polynomial basis functions (instead of Fourier basis functions) -- these could then be the feature maps corresponding to a degree $d$-polynomial kernel.
>
> > Second, how are the hyperparameters for the similarity metric (epsilon and tau) chosen?
>
> While these hyperparameters are indeed important, we found that a large number of configurations ($\tau \in [-1, 0.6]$ and $\epsilon \in [0.9, 2]$) resulted in good performance on the validation set and were easy to train. However, smaller values $\tau$ yielded less interaction sparsity (and therefore less robustness); our model-selection heuristic was therefore to chose $\tau$ as large as possible (and $\epsilon$ as small as possible) without significantly degrading performance on the validation set. The next update will include an ablation on a few aspects of the attention mechanism.
>
> **Continued in part 2.**
>
> ---------- REFERENCES ----------
> - [1] https://people.eecs.berkeley.edu/~brecht/papers/07.rah.rec.nips.pdf
> - [2] https://arxiv.org/abs/2003.08934
> - [3] https://arxiv.org/abs/2006.10739
> - [4] https://arxiv.org/abs/1806.08734
> - [5] https://arxiv.org/abs/1909.10893
> - [6] https://arxiv.org/abs/1909.05215

---

> > ### Author Response · Authors · 2020-11-13
> > **First response to R1 (part 2): on the spatial (and modular) inductive biases**
> >
> > **Continued from part 1 (parent comment)**
> >
> > > Third, from what I can tell, the S2GRU has less capacity than a system that models all of the interactions, such as a global LSTM. If so, then how come the LSTM underperforms on simple tasks such as the bouncing balls? ...
> >
> > As we read from Figure 4 (Performance metrics on OOD one-step forward prediction task...), when both the training set and the test set contain videos with 3 bouncing balls, S2GRUs does not significantly outperform LSTMs and is even outperformed by a modern RNN architecture like Relational Memory Cores (RMCs). However, when the training set contains the same 3 bouncing balls but the test set contains a different number of bouncing balls (1, 2, 4, 5, 6), S2GRUs are able to maintain their performance and generalize out-of-distribution (without additional training) whereas LSTMs and RMCs are adversely affected.
> >
> > This clearly demonstrates that the inductive biases of spatial structure and modularity is indeed helping S2GRU generalize out-of-distribution, whereas models without a strong inductive bias (e.g. LSTMs) generalize well to the data they have seen, but not to novel scenarios. Moreover, we also observe that RIMs, which incorporates the inductive bias of modularity, is able to perform better than LSTMs and RMCs, but not as well as S2GRUs (due to missing spatial structure). Of course, if we were to include sequences with 1, 2, 4, 5, 6 balls in the training dataset, the LSTM should learn to perform better -- nevertheless, in the extreme case of no-additional training, models with appropriate inductive biases outperform those without.
> >
> > > Fourth, are there cases where the spatial structure is not a good inductive bias?
> >
> > This is a great question.
> >
> > One may hypothesize that the spatial structure is not as apparent in the natural language domain. But at the same time, it is known that word-embeddings tend to exhibit rich spatial structure, which in turn S2RMs might be able to exploit.
> >
> > Further, S2RM combines the inductive biases of spatial structure and modularity, implying that even if the spatial structure is not useful, modularity could still be helpful (as is already known from e.g. Appendix F of [5] for NLP transfer-learning tasks).
> >
> > This concludes our first response. We will upload an updated manuscript as soon as possible, but if you have more questions or comments in the mean time, we are eager to discuss!
> >
> > ---------- REFERENCES ----------
> > - [1] https://people.eecs.berkeley.edu/~brecht/papers/07.rah.rec.nips.pdf
> > - [2] https://arxiv.org/abs/2003.08934
> > - [3] https://arxiv.org/abs/2006.10739
> > - [4] https://arxiv.org/abs/1806.08734
> > - [5] https://arxiv.org/abs/1909.10893
> > - [6] https://arxiv.org/abs/1909.05215

---

> > > ### Comment · AnonReviewer1 · 2020-11-25
> > > **Thanks for the clarifications**
> > >
> > > Thanks to the authors for the clarifications. I appreciate the new section (Appendix C.3) on the positional embedding. The choice of hyperparameters still feels a bit vague (e.g. is the range [-1, 0.6] for \tau large or small? and similarly for \epsilon), but that is a minor concern.

---

> > > > ### Author Response · Authors · 2020-11-25
> > > > **Thank you for your response! We have updated the manuscript to include intuition behind how to set tau and epsilon.**
> > > >
> > > > We appreciate your response to our rebuttal and are glad that the additional discussion on the positional encoding was helpful!
> > > >
> > > > > The choice of hyperparameters still feels a bit vague (e.g. is the range [-1, 0.6] for \tau large or small? and similarly for \epsilon)
> > > >
> > > > Thank you for this feedback! We have updated the manuscript to include intuition behind how they should be set (Appendix C.1). In our first comment, we meant that $(\tau, \epsilon) \in [-1, 0.6] \times [0.9, 2]$ results in good results. Here, $\tau$ is the threshold on a dot-product between two unit vectors; accordingly, it should always lie between $[-1, 1]$. Where $Z > 0$, the parameter $\epsilon$ behaves like the reciprocal-width of a gaussian kernel, i.e. small $\epsilon$ results in a wide kernel.
> > > > It can in principle be anywhere between $(0, \infty)$, but setting it too large results in a sharp kernel and permits too little interaction between modules / observations, whereas setting it too small results in a flat kernel, which is detrimental to the propagation of gradients.

---

> ### Author Response · Authors · 2020-11-20
> **Second response to R1: updated manuscript includes a discussion on the positional encoding**
>
> Hello R1, thanks again for your review and your questions! The updated manuscript includes an appendix discussing the details of the positional encoding ([full changelog can be found in this top-level comment](https://openreview.net/forum?id=5l9zj5G7vDY&noteId=PJV-j1Filh)). If there is something you would like us to clarify further, please do not hesitate to let us know!

---

### Author Response · Authors · 2020-11-20
**Change Log**

We are grateful to all reviewers, whose feedback has helped us polish the presentation and solidify our contributions.

### List of Changes (v1)

#### Writing
- We have streamlined the introductory sections; they now require one page less.
- Section 4 now contains more details about the attention mechanism, which we now call _Kernel Modulated Dot Product Attention_, or KMDPA.
- We provide more intuition as to what purpose the components of the input and inter-cell attention mechanisms serve.
- Related work is now discussed in more detail.
- We have tuned the text in experiment section to reflect the new experiments (see below).
- We include a section (Appendix C.3) discussing the positional encoding.
- We discuss limitations and avenues of future research in the concluding section.

#### Experiments
- We include a suite of ablation experiments to examine the importance of various components in the attention mechanisms.
- We include an experiment where we remove modules at test time and measure how the performance degrades.

### List of Changes (v2)
#### Writing
- We include additional discussion building intuition on how to set the parameters $\epsilon$ and $\tau$ (Appendix C.1).

---

### Decision · Program_Chairs · 2021-01-07
**Final Decision**

**Decision:**

Accept (Poster)

**Comment:**

This paper presents a model for dynamical systems with multiple interacting components. Each component is modeled as an RNN, and the interactions between components are functions of their distance in a learned embedding space. It's an interesting idea and well motivated inductive bias. The results were made more compelling with the addition of "ablation" studies during the discussion phase, which showed how various aspects of the model combined to yield the best performance.  Overall, this paper should be of interest to many in the ICLR community working on complex, multi-agent systems.